

# Determining the Daytime Earth Radiative Flux from National Institute of Standards and Technology Advanced Radiometer (NISTAR) Measurements

WENYING SU[1], PATRICK MINNIS[2], LUSHENG LIANG[2], DAVID P. DUDA[2], Konstantin Khlopenkov[2], Mandana M. Thieman[2], Yinan Yu[3], Allan Smith[3], Steven Lorentz[3], Daniel Feldman[4], Francisco P. J. Valero[5]

[1]*Science Directorate, NASA Langley Research Center, Hampton, Virginia*

[2]*Science Systems & Applications, Inc., Hampton, Virginia*

[3]*L-1 Standards and Technology, Inc., New Windsor, Maryland*

[4]*Lawrence Berkeley National Laboratory, MS 84R0171, Berkeley, California*

[5]*Scripps Institute of Oceanography, University of California, San Diego, CA, USA*





## ABSTRACT

The National Institute of Standards and Technology Advanced Radiometer (NISTAR) on-board Deep Space Climate Observatory (DSCOVR) provides continuous full disc global broadband irradiance measurements over most of the sunlit side of the Earth. The three active cavity radiometers measures the total radiant energy from the sun-lit side of the Earth in shortwave (SW, 0.2-4 $\mu$m), total (0.4-100 $\mu$m), and near-infrared (NIR, 0.7-4 $\mu$m) channels. The Level 1 NISTAR dataset provides the filtered radiances (the ratio between irradiance and solid angle). To determine the daytime top-of-atmosphere (TOA) shortwave and long-wave radiative fluxes, the NISTAR measured shortwave radiances must be unfiltered first. An unfiltering algorithm was developed for the NISTAR SW and NIR channels using a spectral radiance data base calculated for typical Earth scenes. The resulting unfiltered NISTAR radiances are then converted to full disk daytime SW and LW flux, by accounting for the anisotropic characteristics of the Earth-reflected and emitted radiances. The anisotropy factors are determined using scene identifications determined from multiple low Earth orbit and geostationary satellites and the angular distribution models (ADMs) developed using data collected by the Clouds and the Earth's Radiant Energy System (CERES). Global annual daytime mean SW fluxes from NISTAR are about 6% greater than those from CERES, and both show strong diurnal variations with daily maximum-minimum differences as great as 20 Wm$^{-2}$ depending on the conditions of the sunlit portion of the Earth. They are also highly correlated, having correlation coefficients of 0.89, indicating that they both capture the diurnal variation. Global annual daytime mean LW fluxes from NISTAR are about 3% greater than those from CERES, but the correlation between them is only about 0.38.



# 1. Introduction

The Earth's climate is determined by the amount and distribution of the incoming solar radiation absorbed and the outgoing longwave radiation (OLR) emitted by the Earth. Satellite observations of Earth Radiation Budget (ERB) provide critical information needed to better understand the driving mechanisms of climate change; the ERB has been monitored from space since the early satellite missions of the late 1950s and the 1960s (House et al. 1986). Currently, the Clouds and the Earth's Radiant Energy System (CERES) instruments (Wielicki et al. 1996; Loeb et al. 2016) have been providing continuous global top-of-atmosphere (TOA) reflected shortwave radiation and OLR since 2000. CERES data have been crucial to advance our understanding of the Earth's energy balance (e.g., Trenberth et al. 2009; Kato et al. 2011; Loeb et al. 2012; Stephens et al. 2012), aerosol direct radiative effects (e.g., Satheesh and Ramanathan 2000; Zhang et al. 2005; Loeb and Manalo-Smith 2005; Su et al. 2013), aerosol-cloud interactions (e.g., Loeb and Schuster 2008; Quaas et al. 2008; Su et al. 2010b), and to evaluate global general circulation models (e.g., Pincus et al. 2008; Su et al. 2010a; Wang and Su 2013; Wild et al. 2013).

The Earth's radiative flux data record is augmented by the launch of the Deep Space Climate Observatory (DSCOVR) on February 11, 2015. DSCOVR is designed to continuously monitor the sunlit side of the Earth, being the first Earth-observing satellite at the Lagrange-1 (L1) point, $\sim$1.5 million km from Earth, where it orbits the Sun at the same rate as the Earth (see Figure 1a). DSCOVR is in an elliptical Lissajous orbit around the L1 point and is not positioned exactly on the Earth-sun line, therefore only about 92$\sim$97% of the sunlit Earth is visible to DSCOVR. As illustrated in Figure 1b, the daytime portion ($A_h$) is not visible to the DSCOVR. Strictly speaking, the measurements from DSCOVR are not truly 'global' daytime measurements. However, for simplicity we refer to them as global daytime measurements. Onboard DSCOVR, the National Institute of Standards and Technology Advanced Radiometer (NISTAR) provides continuous full disc global broadband irradiance measurements over most of the sunlit side of the Earth. Besides NISTAR, DSCOVR also



carries the Earth Polychromatic Imaging Camera (EPIC) which provides 2048 by 2048 pixel imagery 10 to 22 times per day in 10 spectral bands from 317 to 780 nm. On June 8, 2015, more than 100 days after launch, DSCOVR started orbiting around the L1 point.

The NISTAR instrument was designed to measure the global daytime shortwave (SW) and longwave (LW) radiative fluxes. NISTAR measures an irradiance at the L1 point at a small relative azimuth angle, $\phi_o$, which varies from 4° to 15°, as shown in Figure 1a. As such, the radiation it measures comes from the near-backscatter position, which is different from that seen at other satellite positions as indicated in Figure 1a by the varying arrow lengths corresponding to scattering angles, $\Theta_1 - \Theta_3$. Other types of Earth-orbiting satellites view a given spot on the Earth from various scattering angles that vary as a function of local time (e.g., geostationary) or overpass time (e.g., Sun-synchronous). When averaged over the globe, the uncertainties in the anisotropy corrections are mitigated by compensation. That is, any small biases at particular angles are balanced by observations taken at other angles. In contrast, instruments on DSCOVR view every spot on the Earth from a single scattering angle that varies slowly within a small range over the course of the Lissajous orbit. Thus, the correction for anisotropy is critical. The biases in the anisotropy correction for the DSCOVR scattering angle are mitigated and potentially minimized by the wide range of different scene types viewed in a given NISTAR measurement.

Su et al. (2018) described the methodology to derive the global mean daytime shortwave (SW) anisotropic factors by using the CERES angular distribution models (ADMs) and a cloud property composite based on lower Earth orbiting satellite imager retrievals. These SW anisotropic factors were applied to EPIC broadband SW radiances, that were estimated from EPIC narrowband observations based upon narrowband-to-broadband regressions, to derive the global daytime SW fluxes. Daily mean EPIC and CERES SW fluxes calculated using concurrent hours agree with each other to within 2%. They concluded that the SW flux agreement is within the calibration and algorithm uncertainties, which indicates that the method developed to calculate the global anisotropic factors from the CERES ADMs





is robust and that the CERES ADMs accurately account for the Earth's anisotropy in the near-backscatter direction.

In this paper, the same global daytime mean anisotropic factors developed by Su et al. (2018) are applied to the NISTAR measurements to derive the global daytime mean SW and longwave (LW) fluxes. The NISTAR data and the unfiltering algorithms developed for the NISTAR shortwave and near-infrared channels are detailed in section 2. The data and methodology used to derive the global daytime mean anisotropic factors are presented in section 3. Hourly daytime SW and LW fluxes calculated from NISTAR measurements and comparisons with the CERES Synoptic flux products (SYN1deg, Doelling et al. 2013) are detailed in section 4, followed by conclusions and discussions in section 5.

## 2. NISTAR observation

The NISTAR instrument measures Earth irradiance data for an entire hemisphere using active cavity radiometers for three channels: shortwave (SW, 0.2-4.0 $\mu$m), near-infrared (NIR, 0.7-4.0 $\mu$m), and total (0.2-100 $\mu$m). The NISTAR Level 1B (L1B) Earth irradiance data were derived by applying an SI-traceable ground calibration, a phase sensitive demodulation algorithm, and dark offset measurements. These irradiances are reported at the L1 altitude and they are divided by the solid angle ($\Theta$) to provide the respective radiances at the surface ($I$).

Filters are placed in front of the cavity radiometers to measure the energies from the SW and NIR portions of the spectrum. Since no corrections for the impact of filter transmission were applied to the NISTAR L1B data, the SW and NIR radiances from NISTAR must first be unfiltered before they can be used to derive daytime Earth's radiative flux. Here we describe an algorithm to convert measured NISTAR filtered radiances to unfiltered radiances.

Unfiltered SW and NIR radiances are defined as follows:

$$I_u^{band} = \int_{\lambda_1}^{\lambda_2} I_\lambda d\lambda, \tag{1}$$





where 'band' represent either SW or NIR, $\lambda(\mu m)$ is the wavelength, and $I_\lambda$ (Wm$^{-2}$ sr$^{-1}$ $\mu$m$^{-1}$) is the spectral SW radiance. The filtered radiance is the radiation that passes through the spectral filter and is measured by the detector:

$$I_f^{band} = \int_{\lambda_1}^{\lambda_2} S_\lambda^{band} I_\lambda d\lambda, \qquad (2)$$

where $S_\lambda^{band}$ is the spectral transmission function. Figure 2 shows the NISTAR SW and NIR spectral transmission functions. These functions are determined from ground testing done in 1999 and 2010 at the National Institute of Standards and Technology (NIST).

Unfiltered SW and NIR radiances are determined from the filtered radiance measurements as follows:

$$I_u^{sw} = a_0 + a_1(I_f^{sw}) + a_2(I_f^{sw})^2, \qquad (3)$$

$$I_u^{nir} = b_0 + b_1(I_f^{nir}) + b_2(I_f^{nir})^2. \qquad (4)$$

Here $a_0, a_1, a_2, b_0, b_1,$ and $b_2$ are theoretically derived regression coefficients that depend on scene type and Sun-viewing geometry. They are determined from a regression analysis of theoretically derived filtered and unfiltered radiances simulated for typical Earth scenes and the spectral transmission functions shown in Figure 2.

The spectral radiance database is calculated using high-spectral-resolution radiative transfer model (Kato et al. 2002). Unfiltered radiances are determined by integrating spectral radiances over the appropriate wavelength intervals using Gaussian quadrature. Similarly, filtered radiances are computed by integrating over the product of spectral radiance and spectral transmission function. The regression coefficients are derived at 480 angles: 6 solar zenith angles (0.0, 29.0, 41.4, 60.0, 75.5, 85.0 degrees), 8 viewing zenith angles (0, 12, 24, 36, 48, 60, 72, 84 degrees), and 10 relative azimuth angles (0 to 180, at every 20 degrees). For angles between those given above, the regression coefficients are derived by linear interpolation.

The database includes spectral radiances calculated over ocean, land/desert, snow/ice surfaces for clear and cloudy conditions. Table 1 summarizes the cases that are included in





the database, there are a total of 722 clear-sky cases and a total of 1519 cloudy-sky cases

for each Sun-viewing geometry. Regression coefficients are derived based upon the simulated

radiances in this database separately for clear and cloudy conditions for ocean, land/desert,

and snow/ice for each Sun-viewing geometry.

The ratio, $\kappa$, between filtered and unfiltered radiances is calculated for SW and NIR

bands. Table 2 lists the mean and the standard deviation of the ratios at different solar

zenith angles. The ratios for the SW band are extremely stable, varying less than 0.3%

among the scenes and Sun-viewing geometries considered. However, the variability in the

ratios of the NIR band can be as large as 6%. As the NISTAR view always contains clouds,

we choose to use the mean ratios of the cloudy ocean and land cases in Table 2, which

is 0.8690 for the SW band. The estimated uncertainty for the SW band caused by the

unfiltering process is less than 0.1%. The mean ratio for the NIR band is 0.8583, and the

unfiltering uncertainty can be as large as 1∼2%. These mean ratios of the SW and NIR

bands are used to convert the filtered radiances to unfiltered radiances:

$$I_u^{sw} = \frac{I_f^{sw}}{\kappa^{sw}}, \tag{5}$$

$$I_u^{nir} = \frac{I_f^{nir}}{\kappa^{nir}}. \tag{6}$$

Here $I_f^{sw}$ and $I_f^{nir}$ are the filtered radiances directly from the NISTAR L1B data. As there

is no filter placed in front of the total channel, the radiance from the total channel does

not need to be unfiltered. The LW (4-100 $\mu$m) radiance can be derived by subtracting the

unfiltered SW radiance from the total:

$$I_u^{lw} = I^{tot} - I_u^{sw}, \tag{7}$$

The unfiltered radiances ($I_u^{sw}$, $I_u^{lw}$, and $I_u^{nir}$) will be used hereafter to derive the daytime

mean radiative flux. Although NISTAR L1B data provide observations every second, hourly

data (smoothed with 4-hour running mean) are used to derive fluxes because of the level of

noise presented in the measurements (DSCOVR NISTAR data quality report v02).





# 3. Global daytime shortwave and longwave anisotropic factors

To derive the global daytime mean SW and LW fluxes from the NISTAR unfiltered radiances, the anisotropy of the TOA radiance field must be considered. The CERES Edition 4 empirical ADMs and a cloud property composite based upon lower Earth orbit satellite retrievals are used here to estimate the global mean shortwave and longwave anisotropic factors.

*a. CERES ADMs*

The Edition 4 CERES ADMs (Su et al. 2015) are constructed using the CERES observations taken during the rotating azimuth plane (RAP) scan mode. In this mode, the instrument scans in elevation as it rotates in azimuth, thus acquiring radiance measurements from a wide range of viewing combinations. The CERES ADMs are derived for various scene types, which are defined using a combination of variables (e.g., surface type, cloud fraction, cloud optical depth, cloud phase, aerosol optical depth, precipitable water, lapse rate, etc). To provide accurate scene type information within CERES footprints, imager (Moderate Resolution Imaging Spectroradiometer (MODIS) on *Terra* and *Aqua*) cloud and aerosol retrievals (Minnis et al. 2010, 2011) are averaged over CERES footprints by accounting for the CERES point spread function (PSF, Smith 1994) and are used for scene type classification. Over a given scene type ($\chi$), the CERES measured radiances are sorted into discrete angular bins. Averaged radiances ($\hat{I}$) in all angular bins are calculated and all radiances in the upwelling directions are integrated to provide the ADM flux ($\hat{F}$). The ADM anisotropic factors ($R$) for scene type $\chi$ are then calculated as:

$$R(\theta_0, \theta, \phi, \chi) = \frac{\pi \hat{I}(\theta_0, \theta, \phi, \chi)}{\int_0^{2\pi} \int_0^{\frac{\pi}{2}} \hat{I}(\theta_0, \theta, \phi, \chi) cos\theta sin\theta d\theta d\phi} = \frac{\pi \hat{I}(\theta_0, \theta, \phi, \chi)}{\hat{F}(\theta_0, \chi)}, \tag{8}$$



where $\theta_0$ is the solar zenith angle, $\theta$ is the CERES viewing zenith angle, and $\phi$ is the relative azimuth angle between CERES and the solar plane.

*b. EPIC composite data*

As stated in the section above, anisotropy of the radiation field at the TOA was constructed for different scene types, which were defined using many variables including cloud properties such as cloud fraction, cloud optical depth, and cloud phase (Loeb et al. 2005; Su et al. 2015). Although the EPIC L2 cloud product includes threshold-based cloud mask, which identifies the EPIC pixels as high confident clear, low confident clear, high confident cloudy, and low confident cloudy (Yang et al. 2018), the low resolution of EPIC imagery ($24 \times 24$ km$^2$) and its lack of infrared channels diminish its capability to identify clouds and to accurately retrieve cloud properties. As EPIC lacks the channels that are suitable for cloud size and phase retrievals (Meyer et al. 2016), two cloud optical depths are determined assuming the cloud phase is liquid or ice using constant cloud effective radius ($14\mu$m for liquid and $30\mu$m for ice) for cloudy EPIC pixels. These cloud properties are not sufficient to provide the scene type information necessary for ADM selections. Therefore, more accurate cloud property retrievals are needed to provide anisotropy characterizations to convert radiances to fluxes.

To accomplish this, we take advantage of the cloud property retrievals from multiple imagers on low Earth orbit (LEO) satellites and geostationary (GEO) satellites. The LEO satellite imagers include the MODerate-resolution Imaging Spectroradiometer (MODIS) on the Terra and Aqua satellites, the Visible Infrared Imaging Suite(VIIRS) on the Suomi-National Polar-orbiting Partnership satellite, and the Advanced Very High Resolution Radiometer (AVHRR) on the NOAA and MetOps platforms. The GEO imagers are on the Geostationary Operational Environmental Satellites (GOES), the Meteosat series, and Himawari-8 to provide semi-global coverage. All cloud properties were determined using a common set of algorithms, the Satellite ClOud and Radiation Property retrieval System (SatCORPS, Min-



nis et al. 2008b, 2016), based on the CERES cloud detection and retrieval system (Minnis et al. 2008a, 2010, 2011). Cloud properties from these LEO/GEO imagers are optimally merged together to provide a seamless global composite product at 5-km resolution by using an aggregated rating that considers five parameters (nominal satellite resolution, pixel time relative to the EPIC observation time, viewing zenith angle, distance from day/night terminator, and sun glint factor to minimize the usage of data taken in the glint region) and selects the best observation at the time nearest to the EPIC measurements. About 80% of the LEO/GEO satellite overpass times are within 40 minutes of the EPIC measurements, while 96% are within two hours of the EPIC measurements. Most of the regions covered by GEO satellites (between around 50°S and 50°N) have a very small time difference, in the range of ±30 minutes, because the availability of hourly GEO observations. The polar regions are also covered very well by polar orbiters. Thus, larger time differences are generally occurred over the 50° to 70° latitude regions. Given the temporal resolution of the currently available GEO/LEO satellites, this is the best collocation possible for those latitudes. The global composite data are then remapped into the EPIC FOV by convolving the high-resolution cloud properties with the EPIC point spread function (PSF) defined with a half-pixel accuracy to produce the EPIC composite. As the PSF is sampled with half-pixel accuracy, the nominal spacing of the PSF grid is about the same size as in the global composite data. Thus, the accuracy of the cloud fraction in the EPIC composite is not degraded compared to the global composite (Khlopenkov et al. 2017). PSF-weighted averages of radiances and cloud properties are computed separately for each cloud phase, because the LEO/GEO cloud products are retrieved separately for liquid and ice clouds (Minnis et al. 2008b). Ancillary data (i.e. surface type, snow and ice map, skin temperature, precipitable water, etc.) needed for anisotropic factor selections are also included in the EPIC composite. These composite images are produced for each observation time of the EPIC instrument (typically 300 to 600 composites per month). Detailed descriptions of the method and the input used to generate the global and EPIC composites are provided in Khlopenkov et al. (2017).





Figure 3(a) shows an image from EPIC taken on May 15, 2017 at 12:17 UTC, the cor-

responding total cloud fraction (the sum of liquid and ice cloud fractions) from the EPIC

composite is shown in 3(b). The liquid and ice cloud fraction, optical depth, and effective

height are shown in Figure 3(c-h). For this case, most of the clouds are in the liquid phase.

Optically thick liquid clouds with effective heights of 2 to 4 km are observed in the northern

Atlantic ocean and in the Arctic. Ice clouds with effective heights of 8 to 10 km are observed

off the west coast of Africa and Europe.

*c. Calculating global daytime anisotropic factors*

To determine the global daytime mean anisotropic factors, we use the anisotropies char-

acterized in the CERES ADMs and they are selected based upon the scene type information

provided by the EPIC composite for every EPIC FOV. For a given EPIC FOV $(j)$, its

anisotropic factor is determined based upon the Sun-EPIC viewing geometry and the scene

identification information provided by the EPIC composite:

$$R_j(\theta_0, \theta^e, \phi^e, \chi^e) = \frac{\pi \hat{I}_j(\theta_0, \theta^e, \phi^e, \chi^e)}{\hat{F}_j(\theta_0, \chi^e)}, \qquad (9)$$

where $\theta^e$ is the EPIC viewing zenith angle, $\phi^e$ is the relative azimuth angle between EPIC

and the solar plane, and $\chi^e$ is the scene identification from the EPIC composite. To derive

the global mean anisotropic factor, we follow the method developed by Su et al. (2018) and

calculate the global daytime mean ADM radiance as:

$$\overline{\hat{I}} = \frac{\sum_{j=1}^{N} \hat{I}_j(\theta_0, \theta^e, \phi^e, \chi^e)}{N}. \qquad (10)$$

To calculate the global mean ADM flux, we first grid the ADM flux $(\hat{F})$ for each EPIC

pixel into 1° latitude by 1° longitude bins $(\hat{F}(lat, lon))$. These gridded ADM fluxes are then

weighted by *cosine* of latitude to provide the global daytime mean ADM flux:

$$\overline{\hat{F}} = \frac{\sum_{j=1}^{M} \hat{F}_j(lat, lon) cos(lat_j)}{\sum cos(lat_j)}. \qquad (11)$$





The global mean anisotropic factor is calculated as:

$$\overline{R} = \frac{\pi \overline{\hat{I}}}{\overline{\hat{F}}}.$$

(12)

We use $\overline{R_{sw}}$ and $\overline{R_{lw}}$ to denote the mean SW and LW anisotropic factors. The mean SW anisotropic factor is then used to convert the NISTAR SW unfiltered radiance to flux:

$$F_n^{sw} = \frac{\pi I_u^{sw}}{R_{sw}}.$$

(13)

The LW flux is similarly derived from the following:

$$F_n^{lw} = \frac{\pi I_u^{lw}}{R_{lw}}.$$

(14)

Figure 4 shows an example of SW and LW anisotropic factors for every EPIC FOV. The SW anisotropic factors are generally smaller over clear than over cloudy oceanic regions. Over land, however, the SW anisotropic factors are larger over clear regions than over cloudy regions because of the hot spot effect, which leads to anisotropic factors greater than 1.6 over clear land regions at large viewing zenith angles. The LW anisotropic factors show much less variability compare to the SW anisotropic factors, with limb darkening being the dominate feature. The mean SW and LW anisotropic factors for this case are 1.275 and 1.041, respectively.

# 4. NISTAR shortwave and longwave flux

The temporal resolution of the NISTAR Level 1B data is one second, however, meaningful changes in the data only occur over several shutter cycles due to the demodulation algorithm, which includes a box car averaging filter. Following demodulation, significant instrument noise remains. Therefore, further averaging in time over a minimum of 2 hours is recommended to further reduce noise levels (https://eosweb.larc.nasa.gov/project/dscovr /DSCOVR_NISTAR_Data_Quality_Report_V02.pdf). In this study, we use hourly radiances averaged from 4-hour running means as suggested by the NISTAR instrument team. The


hours that are coincident with the EPIC image times are converted to fluxes using the

global anisotropic factors calculated from the EPIC composites. Figure 5 shows the hourly

SW and LW fluxes derived from NISTAR for April (a) and July (b) 2017. For both months,

the SW fluxes fluctuate around 210 $Wm^{-2}$, with the difference between daily maximum and

minimum as large as 30 $Wm^{-2}$. The LW fluxes fluctuate around 260 $Wm^{-2}$, and exhibit

surprisingly large diurnal variations.

These NISTAR fluxes are compared to the CERES Synoptic radiative fluxes and clouds

product (SYN1deg, Doelling et al. 2013), which provides hourly cloud properties and fluxes

for each 1° latitude by 1° longitude. Within the SYN1deg data product, fluxes between

CERES observations are inferred from hourly GEO visible and infrared imager measure-

ments between 60°S and 60°N using observation-based narrowband-to-broadband radiance

and radiance-to-flux conversion algorithms. However, the GEO narrowband channels have

a greater calibration uncertainty than MODIS and CERES. Several procedures are imple-

mented to ensure the consistency between the MODIS-derived and GEO-derived cloud prop-

erties, and between the CERES fluxes and the GEO-based fluxes. These include calibrating

GEO visible radiances against the well-calibrated MODIS 0.65 $\mu$m radiances by ray-matching

MODIS and GEO radiances; applying similar cloud retrieval algorithms to derive cloud prop-

erties from MODIS and GEO observations; and normalizing GEO-based broadband fluxes

to CERES fluxes using coincident measurements. Comparisons with broadband fluxes from

Geostationary Earth Radiation Budget (GERB, Harries et al. 2005) indicate that SYN1deg

hourly fluxes are able to capture the subtle diurnal flux variations. Comparing with the

GERB fluxes, the bias of the SYN SW fluxes is 1.3 $Wm^{-2}$, the monthly regional all-sky SW

flux RMS error is 3.5 $W\,m^{-2}$, and the daily regional all-sky SW flux RMS error is 7.8 $W\,m^{-2}$

(Doelling et al. 2013). These uncertainties could be overestimated, as the GERB domain

has a disproportionate number of strong diurnal cycle regions as compared with the globe.

To account for the missing energy from the daytime portion that is not observed by the

NISTAR ($A_h$ in Figure 1b), the hourly gridded SYN fluxes are integrated by considering



only the grid boxes that are visible to NISTAR to produce the global mean daytime fluxes

that are comparable to those from the NISTAR measurements:

$$\overline{F_{syn}} = \frac{\sum F_j cos(lat_j)\omega_j}{\sum cos(lat_j)\omega_j}. \tag{15}$$

Here $F_j$ is the gridded hourly CERES SYN fluxes, $lat$ is the latitude, and $\omega$ indicates whether

a grid box is visible to NISTAR (=1 when visible, =0 when not visible). Figure 6a) shows

an example of the gridded SYN SW fluxes at 10 UTC on January 1, 2017. SW fluxes

for the daytime grid boxes are shown in color, while all nighttime grid boxes are shown in

white. Figure 6b) shows the area (in red) visible to the NISTAR view, daytime areas of

Scandinavian and South America are not within the NISTAR view and are therefore not

included in the comparison with the NISTAR fluxes.

Figure 7 compares the SW fluxes from NISTAR with those from CERES SYN1deg

product integrated for the NISTAR view (Eq. 19) for April (a) and July (b) 2017. The

CERES SW fluxes oscillate around 200 Wm$^{-2}$ and 195 Wm$^{-2}$ for April and July, whereas

the NISTAR counterparts are about 10 to 20 Wm$^{-2}$ greater. The maxima and minima of

SW fluxes from NISTAR align well with those from CERES, though the differences between

daily maximum and minimum from NISTAR appear to be larger than those from CERES.

The diurnal variations of SW flux derived from EPIC showed a much better agreement with

those from CERES (Su et al. 2018). The exact cause for these larger diurnal variations

from NISTAR SW flux is not known and could be due to onboard data processing. LW

flux comparisons are shown in Figure 8. The daily maximum-minimum LW differences from

CERES are typically less than 15 Wm$^{-2}$ and exhibit small day-to-day and month-to-month

variation. However, the daily maximum-minimum LW differences from NISTAR can vary

from 10 Wm$^{-2}$ to 50 Wm$^{-2}$. These larger than expected variability of NISTAR LW fluxes

are due to the fact that noise and offset variabilities from both the NISTAR total and SW

channel are present in the NISTAR LW radiances. The NISTAR LW fluxes are consistently

greater than CERES LW fluxes by about 10 to 20 Wm$^{-2}$ in April. The LW fluxes agree

better for July, but the NISTAR LW fluxes show larger diurnal variations than the CERES





fluxes.

Figure 9 compares the SW and LW fluxes from CERES SYN1deg product with those
from NISTAR at all coincident hours of 2017. The mean SW fluxes are 204.5 $\mathrm{Wm}^{-2}$ and
217.2 $\mathrm{Wm}^{-2}$, respectively, for CERES and NISTAR, and the RMS error is 14.1 $\mathrm{Wm}^{-2}$ (Fig-
ure 9a). The mean LW fluxes are 246.4 $\mathrm{Wm}^{-2}$ and 252.8 $\mathrm{Wm}^{-2}$ for CERES and NISTAR,
and the RMS error is 10.3 $\mathrm{Wm}^{-2}$ (Figure 9b). Tables 3 and 4 summarize the flux com-
parisons between NISTAR and CERES for all months of 2017. The NISTAR SW fluxes
are consistently greater than those from CERES SYN1deg by about 3.4% to 7.8%, and the
NISTAR LW fluxes are also greater than those from CERES SYN1deg by 1.0% to 5.0%.
Furthermore, the SW fluxes from NISTAR are highly correlated (correlation coefficient of
about 0.89) with those from CERES SYN1deg, but the correlation for the LW fluxes are
rather low (correlation coefficient of about 0.38).

NISTAR fluxes derived at the EPIC image times are averaged into daily means and are
compared with the daily means from CERES SYN1deg using concurrent hours (Figure 10).
The NISTAR SW fluxes are consistently higher than those from CERES by about 10 to 15
$\mathrm{Wm}^{-2}$. CERES SW fluxes show a strong annual cycle, which is driven by the incident solar
radiation that is affected by the Earth-Sun distance. This annual cycle is also evident in the
NISTAR SW fluxes, albeit the fluxes during the period from April to August are flatter than
those from CERES. The NISTAR LW fluxes are greater than those from CERES except
during the boreal summer months, with the largest difference of 10 $\mathrm{Wm}^{-2}$ in February and
the smallest difference of a few $\mathrm{Wm}^{-2}$ during the boreal summer months. The CERES LW
fluxes show an annual cycle of about 10 $\mathrm{Wm}^{-2}$, with the largest LW fluxes occurring during
the boreal summer when the vast land masses of the northern hemisphere are warmer than
during the other seasons. The annual cycle of the NISTAR LW fluxes shows less seasonal
variation. From April to October, the NISTAR LW fluxes oscillate around 255 $\mathrm{Wm}^{-2}$, and
oscillate around 250 $\mathrm{Wm}^{-2}$ for other months. Additionally, the CERES LW fluxes exhibit
much smaller day-to-day variations than their NISTAR counterparts. Note some of the

variations of daily mean fluxes shown in Figure 10 are due to temporal sampling changes when data transmissions encountered difficulties and/or during spacecraft maneuvers.

# 5.  Conclusions and discussions

The SW radiances included in the NISTAR L1B data are filtered radiances and the effect of the filter transmission must be addressed before these measurements can be used to derive any meaningful fluxes. A comprehensive spectral radiance database has been developed to investigate the relationship between filtered and unfiltered radiances using theoretically derived values simulated for typical Earth scenes and the NISTAR spectral transmission functions. The ratio between filtered and unfiltered SW radiances is very stable, varying less than 0.3% for the scenes and the Sun-viewing geometries included in the database. The mean ratio of 0.8690 is used to derive the unfiltered SW radiance from the NISTAR L1B filtered SW radiance measurements.

To convert these unfiltered radiances into fluxes, the anisotropy of the radiance field must be taken into account. We use the scene-type dependent CERES angular distribution models to characterize the global SW and LW anisotropy. These global anisotropies are calculated based upon the anisotropies for each EPIC pixel. To accurately account for the anisotropy for each EPIC pixel, an EPIC composite was developed which includes all information needed for angular distribution model selections. The EPIC composite includes cloud property retrievals from multiple imagers on LEO and GEO satellites. Cloud properties from these LEO and GEO imagers are optimally merged together to provide a global composite product at 5-km resolution by using an aggregated rating that considers several factors and selects the best observation at the time nearest to the EPIC measurements. The global composite data are then remapped into the EPIC FOV by convolving the high-resolution cloud properties with the EPIC PSF to produce the EPIC composite. PSF-weighted averages of radiances and cloud properties are computed separately for each cloud phase, and ancillary data needed





for anisotropic factor selections are also included in the EPIC composite.

These global anisotropies are applied to the NISTAR radiances to produce the global

daytime SW and LW fluxes and they are validated against the CERES Synoptic 1° latitude

by 1° longitude flux product. Only the grid boxes that are visible to the NISTAR view

are integrated to produce the global mean daytime fluxes that are comparable to the fluxes

from the NISTAR measurements. The NISTAR SW fluxes are consistently greater than

those from CERES SYN1deg by 10 Wm$^{-2}$ to 15 Wm$^{-2}$ (3.3% to 7.8%), but these two SW

flux datasets are highly correlated indicating that the diurnal and seasonal variations of

the SW fluxes are fairly similar for both of them. The NISTAR LW fluxes are also greater

than those from CERES SYN1deg, but the magnitude of the difference has larger month-

to-month variations than that for the SW fluxes. The largest difference of about 10 Wm$^{-2}$

($\sim$9%) occurred in January 2017 and the smallest difference of about $\sim$2 Wm$^{-2}$ ($\sim$1%)

occurred during the boreal summer months. Furthermore, the NISTAR LW fluxes have very

low correlations with the CERES LW fluxes. NISTAR LW fluxes exhibit a nearly flat annual

variation, whereas the CERES LW fluxes exhibit a distinct annual cycle with the highest

LW flux occurs in July when the vast northern hemisphere land masses are warmest. The

NISTAR LW fluxes also exhibit unrealistically large day-to-day variations.

The SW flux discrepancy between NISTAR and CERES is caused by: 1) CERES instru-

ment calibration uncertainty, 2) CERES flux algorithm uncertainty, 3) NISTAR instrument

measurement uncertainty, and 4) NISTAR flux algorithm uncertainty. The CERES SW chan-

nel calibration uncertainty is 1% (Loeb et al. 2018), which corresponds to about 2.1 Wm$^{-2}$

for daytime mean SW fluxes. The CERES algorithm uncertainty includes radiance-to-flux

conversion error, which is 1.0 Wm$^{-2}$ according to Su et al. (2015), and diurnal correction un-

certainty, which is estimated to be 1.9 Wm$^{-2}$ when Terra and Aqua are combined (Loeb et al.

2018). The NISTAR SW channel measurement uncertainty is 2.1%, which corresponds to

4.4 Wm$^{-2}$. The NISTAR algorithm uncertainty is essentially the radiance-to-flux conversion

error. The estimation of this error source is not readily available given the unique NISTAR



viewing perspective. However, if we assume the discrepancy between EPIC derived SW flux

and CERES SW flux (Su et al. 2018) is also from uncertainty sources 1) and 2) listed above,

plus the EPIC calibration, narrowband-to-broadband conversion, and radiance-to-flux con-

version for EPIC, then we can deduce that the radiance-to-flux conversion uncertainty for

the NISTAR viewing geometry should be less than 2 $Wm^{-2}$. Thus the total difference ex-

pected from these uncertainty sources should be $(2.1^2 + 1.9^2 + 1.0^2 + 4.4^2 + 2.0^2)^{1/2} = 5.7$

$Wm^{-2}$.

Similarly, the LW flux discrepancy between NISTAR and CERES is due to the same

sources of error. The CERES LW channel calibration uncertainty is 1.8 $Wm^{-2}$. The CERES

LW radiance-to-flux conversion error is about 0.75 $Wm^{-2}$(Su et al. 2015), and diurnal cor-

rection uncertainty is estimated to be 2.2 $Wm^{-2}$ (Loeb et al. 2018). However, the CERES

LW ADMs were developed without taking the relative azimuth angle into consideration,

which has little impact on the CERES LW flux accuracy because of its Sun-synchronous

orbit. Given that the NISTAR only views the Earth from the backscattering angles, the LW

flux uncertainty due to radiance-to-flux conversion could be larger for the clear-sky foot-

prints (Minnis et al. 2004). As the clear-sky occurrences are small at the EPIC footprint

size level, our best guesstimate of this uncertainty is no more than 0.4 $Wm^{-2}$. The cali-

bration uncertainty for NISTAR LW is deduced from the calibration uncertainties of total

and SW channels. The total channel calibration uncertainty is 1.5%, which is about 6.8

$Wm^{-2}$ assuming the total radiative energy of 450 $Wm^{-2}$. The SW channel measurement

uncertainty is 4.4 $Wm^{-2}$. The resulting LW channel measurement uncertainty is thus equal

to $(6.8^2 + 4.4^2)^{1/2} = 8.1$ $Wm^{-2})$. Although no direct estimation of the radiance-to-flux con-

version uncertainty for LW is available, we do not expect that it exceeds its SW counterpart

of 2.0 $Wm^{-2}$. Thus the total difference expected from these uncertainty sources should be

$(1.8^2 + 0.75^2 + 0.4^2 + 2.2^2 + 8.1^2 + 2.0^2)^{1/2} = 8.9Wm^{-2}$.

The uncertainty sources listed above can explain part of the SW flux differences and

all of the LW flux differences between CERES and NISTAR. The error sources related to



NISTAR are preliminary and are under careful evaluation. Although the LW flux differences between CERES and NISTAR are within the uncertainty estimation, the correlation between NISTAR and CERES is rather low, about 0.38. This is because the NISTAR LW radiance is derived as the difference between total channel radiance and SW channel radiance, thus noise and offset variability of both the NISTAR total and SW channels are present in the NISTAR LW fluxes. As a result, more variability is expected in the LW data which leads to the low correlation. The diurnal variations of the SW and LW fluxes from both NISTAR and CERES SYN1deg will be compared with the high-temporal resolution model outputs from the Coupled Model Intercomparison Project.

*Acknowledgments.*

This research was supported by the NASA DSCOVR project. The CERES data were obtained from the NASA Langley Atmospheric Science Data Center at https://eosweb.larc.nasa.gov/project/ceres/ssf_terra-fm1_ed4a_table(ssf_aqua-fm3_ed4a_table). The data used to produce the figures and tables in this paper are available to readers upon request. We thank Szedung Sun-Mack, Rabindra Palikonda, and Kristopher Bedka for processing the LEO/GEO data used to construct the EPIC composite.



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



# List of Tables



TABLE 1. Summary of the cases included in the spectral radiance database. AOD is for aerosol optical depth, COD is for cloud optical depth.

| | | Clear | | |
|---|---|---|---|---|
| | AOD | Aerosol type | Surface | |
| Ocean | 8 | 6 | 4 | |
| Land | 8 | 4 | 15 | |
| Snow | 5 | 2 | 5 | |
| | | Cloudy | | |
| | COD | Cloud type | Surface | Atmosphere |
| Ocean | 7 | 4 liquid and 3 ice | 4 | 4 |
| Land | 7 | 4 liquid and 3 ice | 15 | 1 |

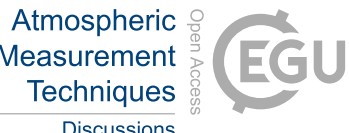

TABLE 2. Mean ratio and standard deviation (in parenthesis) of filtered radiance to unfiltered radiance for SW and NIR bands over different scene types.

| | SW ratio (standard deviation × 1000) | | | | | |
|---|---|---|---|---|---|---|
| | 0.0 | 29.0 | 41.4 | 60.0 | 75.5 | 85.0 |
| Clear Ocean | 0.8659(1.0) | 0.8660(1.0) | 0.8661(1.1) | 0.8664(1.2) | 0.8669(1.0) | 0.8674(0.8) |
| Clear Land | 0.8694(0.6) | 0.8693(0.6) | 0.8692(0.6) | 0.8690(0.5) | 0.8687(0.5) | 0.8685(0.8) |
| Clear Snow | 0.8689(0.1) | 0.8689(0.1) | 0.8689(0.2) | 0.8688(0.2) | 0.8688(0.3) | 0.8687(0.4) |
| Cld Ocean | 0.8687(1.0) | 0.8687(1.0) | 0.8688(0.9) | 0.8688(0.8) | 0.8688(0.7) | 0.8687(0.6) |
| Cld Land | 0.8694(0.4) | 0.8693(0.3) | 0.8693(0.3) | 0.8692(0.3) | 0.8690(0.4) | 0.8689(0.5) |
| | NIR ratio (standard deviation × 1000) | | | | | |
| | 0.0 | 29.0 | 41.4 | 60.0 | 75.5 | 85.0 |
| Clear Ocean | 0.8293(23.1) | 0.8270(24.0) | 0.8253(25.5) | 0.8235(28.3) | 0.8238(28.4) | 0.8229(26.4) |
| Clear Land | 0.8790(9.6) | 0.8777(10.4) | 0.8764(10.7) | 0.8730(10.8) | 0.8663(10.1) | 0.8501(12.4) |
| Clear Snow | 0.8360(1.7) | 0.8360(1.8) | 0.8361(1.9) | 0.8363(2.1) | 0.8370(2.8) | 0.8365(6.0) |
| Cld Ocean | 0.8557(3.2) | 0.8555(2.6) | 0.8562(2.4) | 0.8567(3.1) | 0.8565(4.4) | 0.8539(7.9) |
| Cld Land | 0.8627(8.2) | 0.8624(7.8) | 0.8621(7.3) | 0.8613(6.2) | 0.8598(4.8) | 0.8566(6.2) |





TABLE 3. SW flux comparisons between NISTAR and CERES SYN1deg for all coincident observations of 2017. $F_n$ is the NISTAR flux (in Wm$^{-2}$), $F_s$ is the SYN flux (in Wm$^{-2}$), and $\frac{F_n - F_s}{F_s}$ is the relative difference between them (in %).

|  | Jan | Feb | Mar | Apr | May | Jun | Jul | Aug | Sep | Oct | Nov | Dec |
|---|---|---|---|---|---|---|---|---|---|---|---|---|
| $F_s$ | — | 210.3 | 205.1 | 201.9 | 201.4 | 198.8 | 194.5 | 195.0 | 199.9 | 210.3 | 222.3 | 228.5 |
| $F_n$ | — | 217.5 | 214.3 | 210.4 | 213.7 | 214.3 | 209.5 | 208.4 | 211.1 | 224.1 | 236.1 | 240.5 |
| $\frac{F_n - F_s}{F_s}$ | — | 3.4 | 4.5 | 4.2 | 6.1 | 7.8 | 7.7 | 6.9 | 5.6 | 6.6 | 6.2 | 5.3 |





TABLE 4. LW flux comparisons between NISTAR and CERES SYN1deg for all coincident observations of 2017. $F_n$ is the NISTAR flux (in Wm$^{-2}$), $F_s$ is the SYN flux (in Wm$^{-2}$), and $\frac{F_n - F_s}{F_s}$ is the relative difference between them (in %).

|  | Jan | Feb | Mar | Apr | May | Jun | Jul | Aug | Sep | Oct | Nov | Dec |
|---|---|---|---|---|---|---|---|---|---|---|---|---|
| $F_s$ | — | 242.3 | 242.0 | 244.0 | 247.6 | 250.1 | 251.6 | 249.3 | 246.1 | 243.2 | 240.1 | 241.3 |
| $F_n$ | — | 251.5 | 246.3 | 256.1 | 254.0 | 253.4 | 254.0 | 251.5 | 253.8 | 251.8 | 248.8 | 251.6 |
| $\frac{F_n - F_s}{F_s}$ | — | 3.8 | 1.8 | 5.0 | 2.6 | 1.3 | 1.0 | 0.9 | 3.1 | 3.5 | 3.6 | 4.3 |



# List of Figures



10    Daily mean SW flux (a) and LW flux (b) comparisons between CERES SYN1deg

(blue) and NISTAR (red) for 2017.                                          40

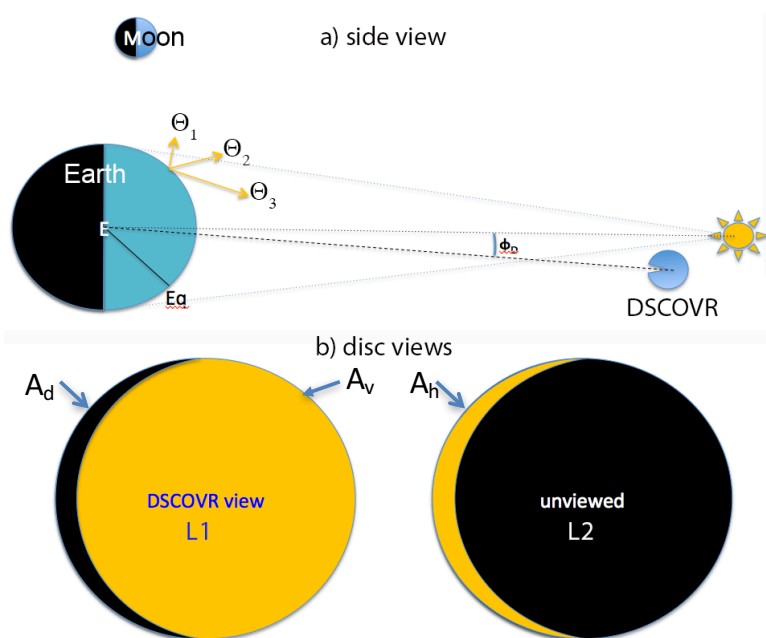

FIG. 1. Schematic of a) Earth-Sun-DSCOVR geometry and b) Earth disc that are visible to the L1 DSCOVR view (left with an area fraction of $A_t$) and to the L2 view (right). The golden area on the left shows the daytime area fraction ($A_v$) that are visible to DSCOVR, the black area on the left shows the night portion ($A_d$) that are within the DSCOVR view, and the golden area on the right is the daytime portion ($A_h$) missed by the DSCOVR. Not to scale.



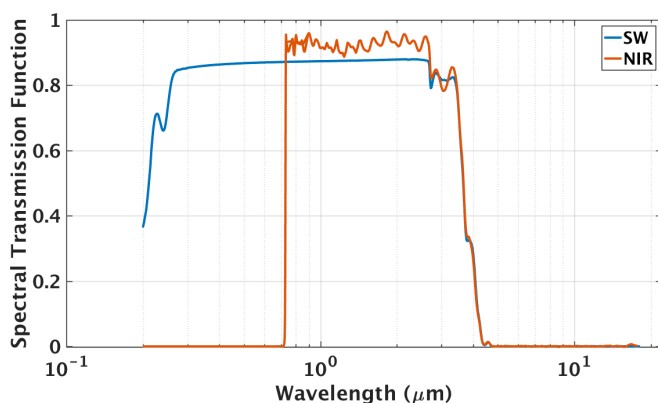

FIG. 2. NISTAR SW and NIR spectral transmission function.

 Atmospheric
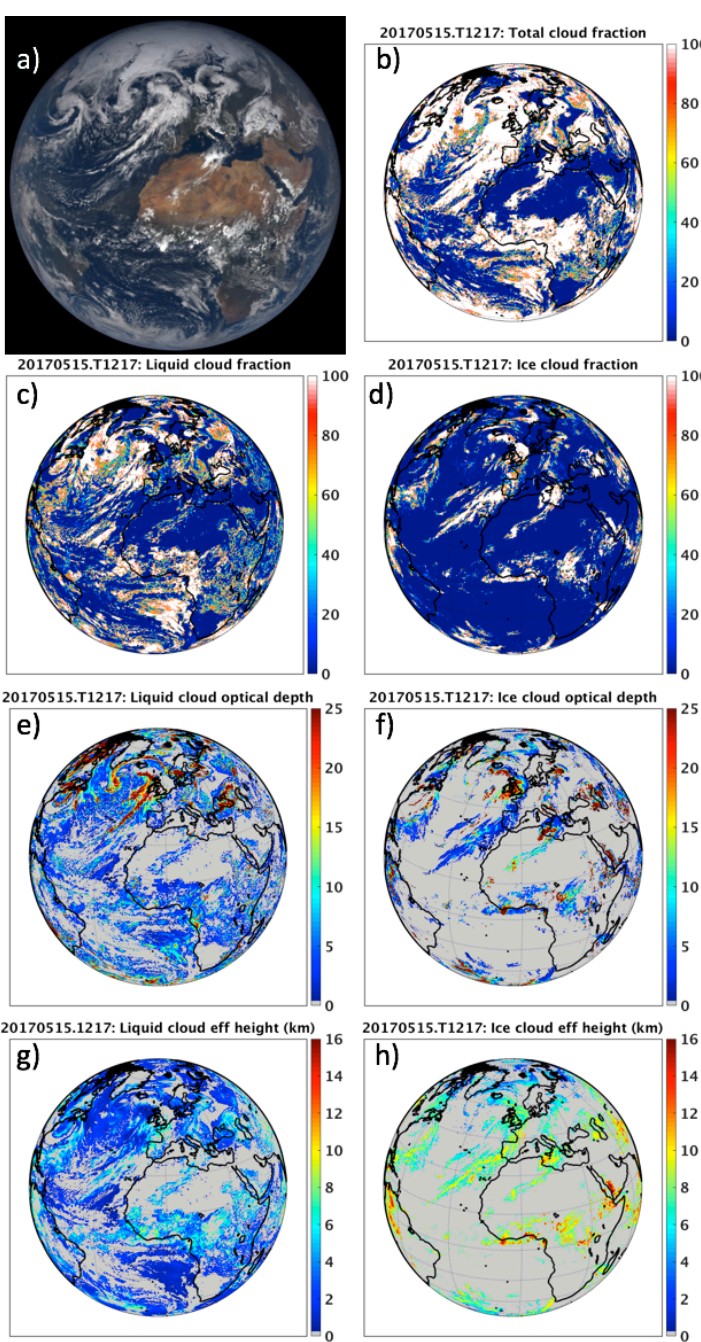

FIG. 3. EPIC RGB image for May 15, 2017 at 12:17 UTC (a), and the corresponding total cloud fraction (b, in %). Liquid and ice cloud fractions are shown in (c) and (d), liquid and ice cloud optical depths are shown in (e) and (f), and liquid and ice cloud effective height (in $km$) are shown in (g) and (h). (b) to (h) are all derived from the EPIC composite.

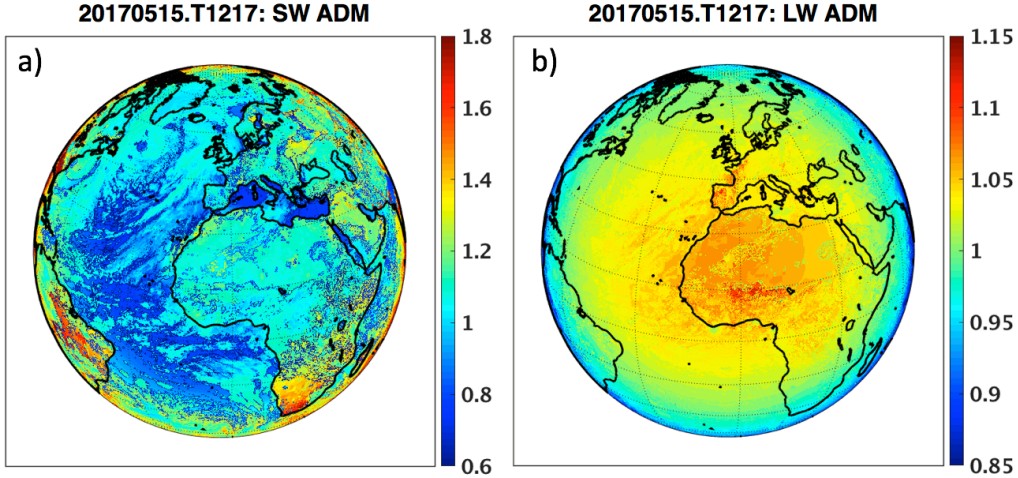

Fig. 4. SW anisotropic factors (a) and LW anisotropic factors (b) derived from the CERES ADMs using the EPIC composite for scene identification for May 15, 2017 at 12:17 UTC.



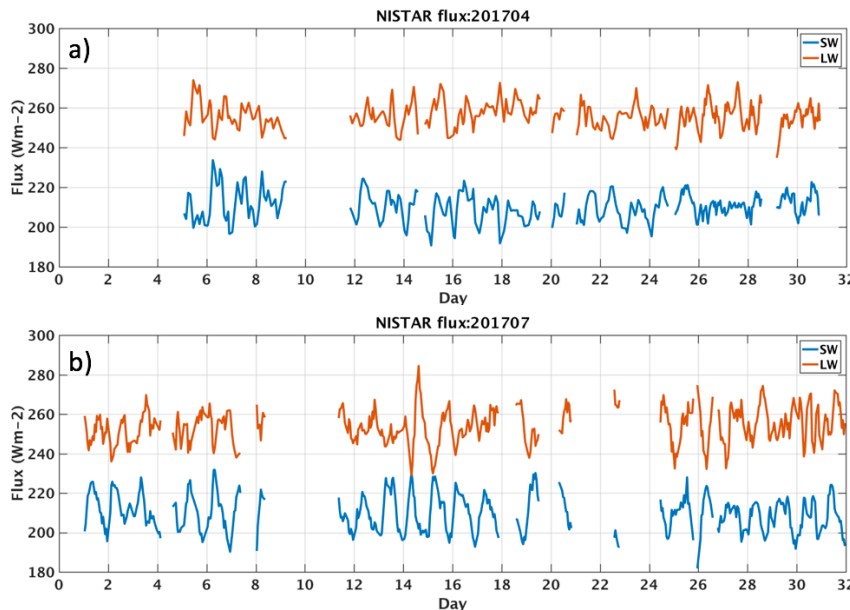

FIG. 5. SW flux (blue) and LW flux (red) derived from NISTAR measurements for April
(a) and July (b), 2017.



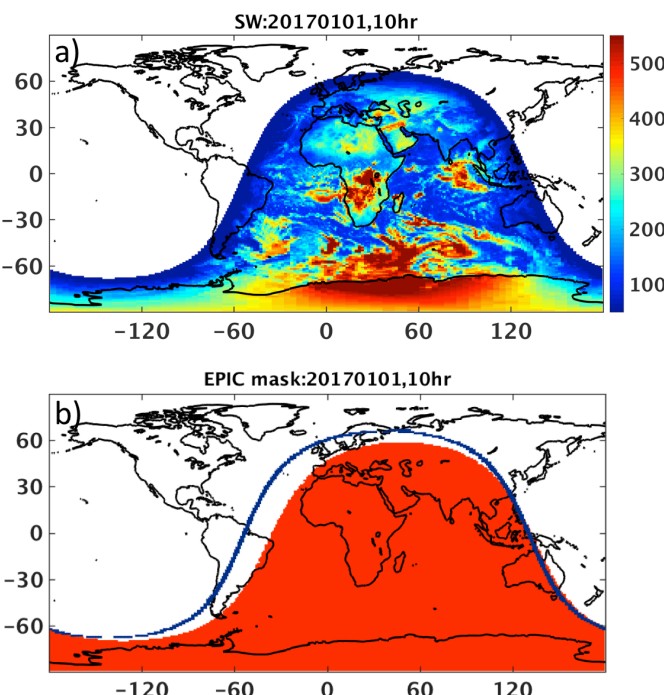

FIG. 6. An example of the daytime SW flux distributions from CERES SYN1deg product at 10 UTC on January 1, 2017 (a), and the corresponding areas (in red) that are visible to EPIC and the terminator boundary (in blue) (b).

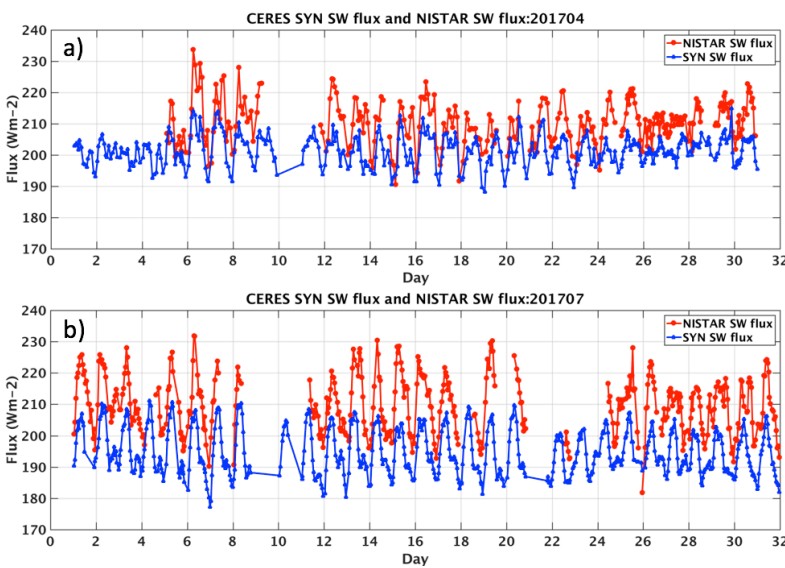

FIG. 7. SW flux (in Wm$^{-2}$) comparisons between NISTAR and CERES SYN for April (a) and July (b) 2017.

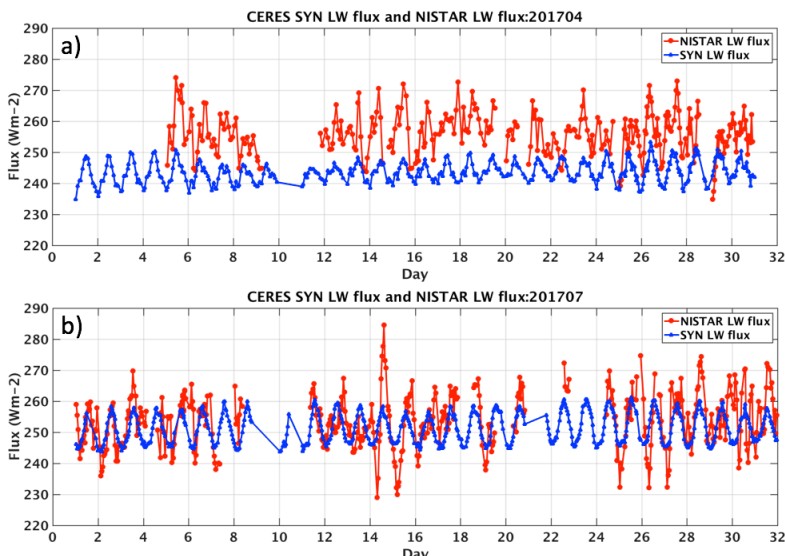

FIG. 8. LW flux (in Wm$^{-2}$) comparisons between NISTAR and CERES SYN for April (a) and July (b) 2017.



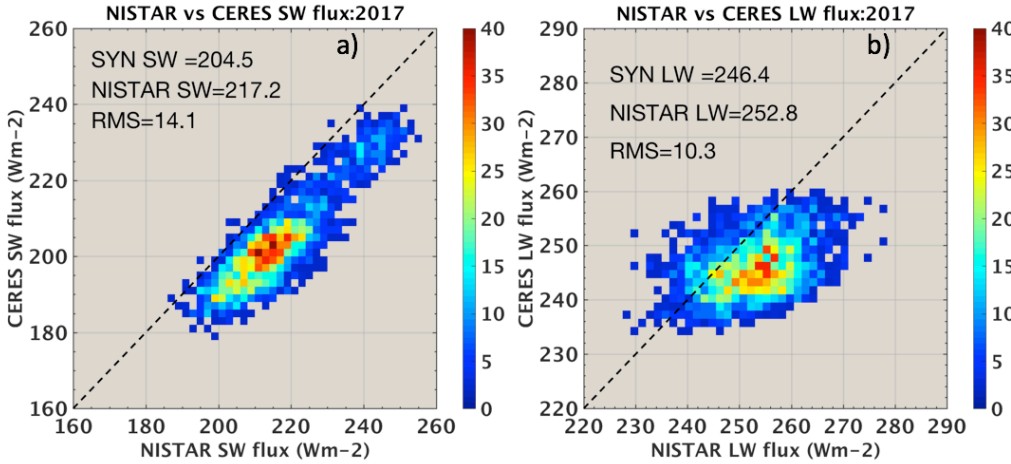

FIG. 9. Comparison of coincident hourly SW and LW fluxes from NISTAR and CERES SYN1deg for 2017. Color bar indicates the number of occurrence.

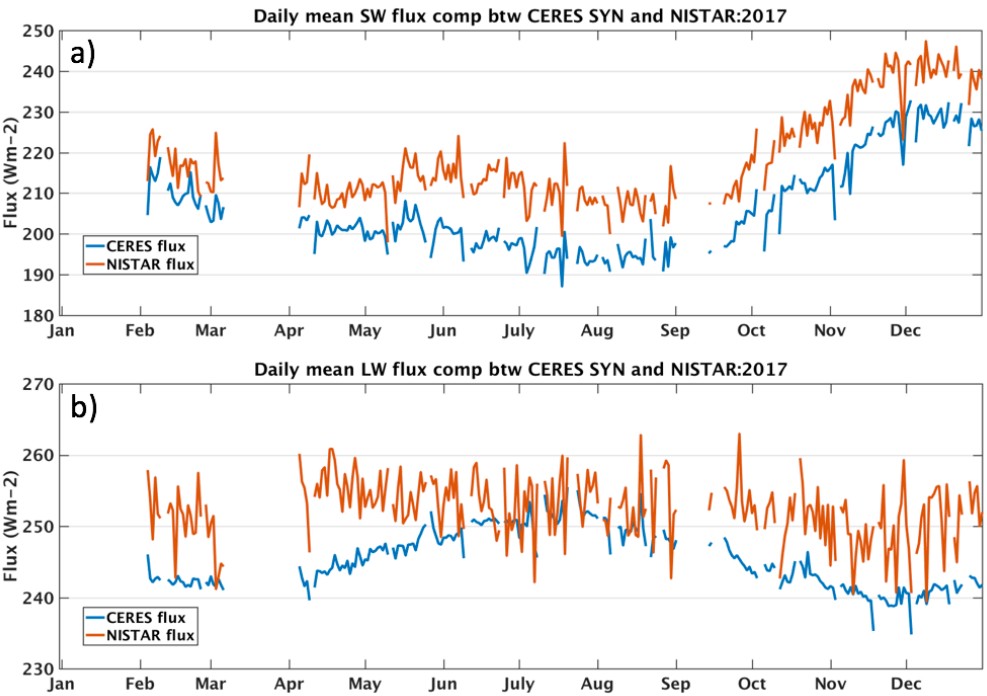

FIG. 10. Daily mean SW flux (a) and LW flux (b) comparisons between CERES SYN1deg (blue) and NISTAR (red) for 2017.