# Peer review of "Determining the Daytime Earth Radiative Flux from National Institute of Standards and Technology Advanced Radiometer (NISTAR) Measurements"

_Atmospheric Measurement Techniques, 2019_

## Referee Comment (RC1) · Anonymous Referee #1 · 18 Aug 2019

11.     In ERB calibration your definition of filtered radiance as IRRADI-ANCE/SOLIDANGLE is only true if the instrument has a completely flat spectral response, which from fig 2 is certainly not the case for NISTAR SW & NIR channels.

113-132.  This is based on CERES unfiltering of Loeb et al 2001 I assume (where they are also labelled Eqns 3 & 4 at https://journals.ametsoc.org/doi/pdf/10.1175/1520-0450%282001%29040%3C0822%3ADOURFT%3E2.0.CO%3B2 ).  Is it completely identical to CERES using the same decades old CERES radiative transfer database? This might be important to briefly mention as it could help eliminate mere inversion biases when you compare to CERES later in the paper.

[Figure]

142. This is confusing, although I accept it probably amounts to same thing, are look up tables of a & b values (Eqns 3 & 4) or a table of kappa ratios actually used? Only if both techniques are used separately should there be 4 rather than just 2 eqns?

143. How are spectrally dependent changes to the transmission of the quartz filter due to outgassing contamination measured after launch and throughout the mission?

144. What about quartz filter leakage? Are you using the NIR channel for that somehow (similar to Loeb et al 2001 above)?

146. Are you sure no unfiltering of the total channel is required, if so how? Was its spectral response measured to be certain? How are you certain no changes to the effective gains of the cavity channels due to electronics radiation exposure are occurring? I'm assuming you do not have onboard blackbodies?

147. How are the SW and Total channels balanced in the solar region as in Kratz et al 2002? (https://agupubs.onlinelibrary.wiley.com/doi/epdf/10.1029/2001JD001170)

261. what is a shutter cycle? Why is a boxcar filter used in the demodulation algorithm? Are details of these processes important?

264. Recommended based on what, the URL does not work?

266. Why 4 hour "running means" and how is this different from a 4-hour wide boxcar filter from the terminology you used earlier? Does this mean a 4-hour running mean is taken of the boxcar filtered then 2-hour averaged data? Why are the 4 hour means suggested by the NISTAR instrument team?

286. These GERB comparisons need a reference.

311. Why does the onboard data processing cause this?

315. How are the offsets countered, space looks?

332. With as few as 10 EPIC results per day are these always equally spaced in time?

If not, could this not lead to biases?

338. So it seems the LW difference is greatest in Northern Hemisphere Winter, when more ocean is observed? This may be a calibration artifact or error in knowledge of the NISTAR SW channel for the UV region. As per the point above for line 147, how are you balancing SW and Total channels to assure accurate LW in daylight?

352. "A comprehensive spectral database has been developed", so is it different from that used by CERES?

355. So is a constant of 0.8690 used for all NISTAR unfiltering? Unfiltering of LEO scenes varies greatly by several percent especially for ocean scenes etc. So, it seems a value of 0.3% difference for primarily land vs Pacific Ocean scenes would vary more (and maybe adds to your seasonal cycle). What results lead to the 0.3% conclusion and did you try a scene by scene unfiltering?

370. Is this the PSF of the EPIC telescope separate from its array of detectors? How was it measured?

388. Again, this could be due to a constant unfiltering factor?

392. Loeb et al 2018 only quotes the 1% accuracy figure as do you, please provide a peer reviewed SI traceable reference.

396. Please give a peer reviewed reference for the 2.1% NISTAR SW accuracy figure.

404. With so many error sources not well known it is wrong to simply add them all in quadrature, which assumes they are all random and independent. A more sophisticated error analysis is needed.

407. The 1.8Wmˆ2 accuracy for CERES LW applies for nighttime LW only. During the day which is always the case for NISTAR it is less accurate. This is because it requires the earlier discussed balancing of the SW and Total channel which if done wrong can result in measuring the Earth warmer at night than during the day for example (see

Fig11b, Page 14 at https://journals.ametsoc.org/doi/pdf/10.1175/2010JTECHA1521.1 ). Hence for NISTAR which only views day LW, this is an important consideration.

415. Guesstimate? This is most unsatisfactory for any science paper, let alone one on climate measurements. Please do better.

423. Again, adding in quadrature for so many uncertain, often modelling terms is not acceptable. For example, consider how the error in knowledge of SW vs Total solar response could be systematic because of an error in the ground lab, it will partly cancel in the Total – SW subtraction.

428. This is true, in addition to the above-mentioned systematic nature of SW and Total errors not considered in your quadrature additions. A more sophisticated analysis is needed.

Overall this paper has merit but needs work to fill in the blanks on some of the processes/references used. The large differences of NISTAR from CERES appears strange and would seem at first look to be largely from algorithm errors. I feel this could be acceptable being a new measurement, but needs to be stated more clearly in the paper as such. The use of constant SW unfiltering also raises concern and leads to the possibility it is a cause of the larger than expected seasonal cycles, but more investigation is needed. Also some insight in the introduction into the purpose of NISTAR would be good, such as giving illustration if and how it complements the climate observing system discussed by Weilicki et al 2013 (https://journals.ametsoc.org/doi/pdf/10.1175/BAMS-D-12-00149.1 ).

In summary, this paper could become suitable for publication, given more work, research and additions that address the points above. It should then be re-considered under peer review.
* * *

---

## Referee Comment (RC2) · Anonymous Referee #3 · 19 Aug 2019

General comments:

This paper presents the scheme and algorithm of deriving TOA SW/LW flux from NISTAR measurements and comparison also made with the corresponding results derived from CERES. I am impressed by the detailed and clear description of the algorithms. The paper is very well written and relevant to the community. I recommend publication after addressing the minor issues listed bellowed. It doesn't seem that the uncertainties in the algorithms would give a consistent bias seeing in the differences between NISTAR and CERES. Has there been analysis with the NISTAR instrument measurements and calibration? The low correlation between NISTAR LW flux and that

of CERES is puzzling. To bypass the potential uncertainties in part of the algorithms, it may be useful to look at the correlation between the NISTAR LW radiances and the CERES flux to see if they are correlated at all.

Specific comments:

Page 5 and 6: The authors have derived the regression equations for the unfiltered radiances (Eq 3 and 4); what is the reason for using the less accurate ratio method (Eq. 5 and 6)?

Page 14: How are the portion of the Earth not visible to NISTAR decided? Also, similar to NISTAR missing some of the daytime portion of the Earth, it must be seeing part of the night time side of the Earth. Are these taking into account for the longwave calculations?

---

## Referee Comment (RC3) · Anonymous Referee #2 · 22 Aug 2019

Review on "*Determining the Daytime Earth Radiative Flux from National Institute of Standards and Technology Advanced Radiometer (NISTAR) Measurements*" by Su et al.

This paper documents the methodology to derive the broadband radiative flux from the measurements of the NISTAR instrument onboard of the DSCOVR mission. Some preliminary results based on this method are compared with the well-developed CERES data. The SW fluxes derived from the NISTAR compares reasonably well with CERES, but the LW fluxes from NISTAR have a systematic bias and low correlation coefficient when benchmarked with CERES.

The topic of this paper is important and suitable for AMT. The paper is well organized. However, the paper lacks some important technical details about the instrument and the methodology, as well as the author's opinion about the usefulness of the NISTAR product. In my view, some significant revisions are needed before the paper can be accepted for publication. Below is a list of questions and concerns I have.

1) The parameterization scheme described in Section 2 to obtain unfiltered radiance from observed filtered radiance is confusing. Up to line 132, the method seems to be based on the polynomial parameterization scheme in Eqs (3) and (4). But then it suddenly changed to the simply ratio-based parameterization in Eqs. (5) and (6). Why are there two types of parameterization? Which one is used?

2) What is the FOV size of the NISTAR instrument? Does it observe the earth pixel by pixel (similar to EPIC) or as a whole? Does its FOV include some cosmic background and, if so, how is that treated?

3) Within its FOV, does the NISTAR instrument response to the radiance from different locations and angles equally? In other words, do the radiances from the edge of the earth disc have the same weighting as those from the center of the disc?

4) It is stated that "*The biases in the anisotropy correction for the DSCOVR scattering angle are mitigated and potentially minimized by the wide range of different scene 71 types viewed in a given NISTAR measurement.*" Some references are needed to support it.

5) In Su et al. (2018), a similar method is used to derive the fluxes from EPIC measurements. One of the byproduct from this EPIC-based method is the "global day-time mean SW radiance" $\overline{I_{bb}}$. Is it something directly comparable to the observation of NISTAR instrument? If so, some comparisons should be made because both EPIC and NISTAR have the similar sun-satellite geometry.

6) I have several questions about the method described in Section 3c. First of all, what is the theoretical based for Eqs 9~ 11? If my understanding is correct, the global mean SW flux is $F = \iint_{sunlit} \frac{I[\theta_0(r),\theta^e(r),\phi^e(r),\chi(r)]}{R(\theta_0,\theta^e,\phi^e\chi)} d^2r$, where r denotes a point on earth. But this is not equal to $\frac{\iint_{sublit} I[\theta_0(r),\theta^e(r),\phi^e(r),\chi(r)]d^2r}{\iint_{sublit} R(\theta_0,\theta^e,\phi^e\chi)d^2r}$. More detailed mathematically derivations are needed here. Secondly, one might ask if a global mean anisotropic factor is even physically meaningful? The average is over a large range of viewing angles and scene

types. Does the result have any physical meaning? Moreover, are the angular and spectral averaging independent and can be treated independently? The derivations in Section 3c seem to suggest they are independent, but this is not obvious to me. Some clarification is needed.

7) This paper only shows "how to do it" but does not explain "why to do it" other than it can be done. I understand that this paper is to document the method used to derive the flux from the radiance observations of NISTAR. But I think in addition to the technical details the reader would apricate some insights and opinions from the authors about the usefulness of the product. We already have the state-of-the-art CERES flux product and in Su et al. (2018) flux product has also been developed. What is new/novel/important about the NISTAR flux product other than the fact it can be done? What kind of applications can this product be used for? Some discussions about these important questions should be added to the abstract and conclusion parts.

---

## Referee Comment (RC4) · Anonymous Referee #4 · 20 Sep 2019

Review of "Determining the Daytime Earth Radiative Flux from National Institute of Standards and Technology Advanced Radiometer (NISTAR) Measurements" by Su et al. 2018

General comments:

This manuscript derives sunlit side of the Earth's radiation budget (SW and LW) from a single pixel measurement of NISTAR instrument on board the DSCOVR mission and compares with the radiation fluxes derived from the CERES measurements. This is a very interesting and important work as the Earth's radiation budget has been so far solely measured by the ERBE/CERES project and there are very little independent

and direct measurements of these important quantities. This work builds upon many previous works the team has been working on for many years including narrowband-broadband conversion, ADM, GEO/LEO composite cloud products etc. The paper is well written and structured. I do have some questions and suggestions regarding the derivation of global ADM and evaluation of each components of the fluxes.

Specific comments:

Line 98: What is the uncertainty level of NISTAR L1B radiance? What kind of calibration procedures have been used to produce the L1B radiance? You have discussed some of the issues later in the paper but it's worthwhile to have a paragraph to discuss the NISTAR at the beginning of the paper. NISTAR provides a completely different methodology of estimating the earth's radiation budget and independent check of Earth's radiation budget created from CERES measurements, the difference found in this article is very serious and should be adequately explained. NISTAR's absolute calibration and uncertainty is of fundamental importance, otherwise the readers would question the well-established CERES products.

Line 147. The conversion from filtered to unfiltered radiances used the ratio derived from model simulation data using eq 5 and 6. Why not using the regression (3) and (4)? The regression indicates the ratio could not be constant because it's a quadratic function and has an offset. It's justified to use a constant ratio between the two if the ratio varies little as for the SW band, but a constant ratio for NIR would introduce an unnecessary source of error (1~2%) for the NIR and I don't see why you should abandon the regression.

Line 152: Did you use NIR in this work? If not, could you explain why NISTAR takes the NIR measurement?

Line 187: EPIC images have 8x8 km2 resolution at nadir and are 1/cos(vza) larger at larger view zenith angles. The EPIC cloud products are retrieved at its native resolution with (2014x2014) pixels in a granule. Some channels have degraded into 1024x1024

for downlink but reversed to 2014x2014 afterwards.

Equation (9) and (11), Ij and Fj seem to refer to radiance and flux in each EPIC composite pixel. Do you actually use those in the mean ADM calculations? If yes, did you use the EPIC measured narrowband radiances to compute the broadband radiance and flux for each pixel? Why did you grid the fluxes into 1x1 grid boxes and not the radiances? The global mean flux is computed from Eq. 11 to take care of different sizes of grids in each latitude. If you grid the radiance, then you would compute the mean radiance the same fashion as the flux. Otherwise, if you average the radiance from each pixel directly, then you would also have to consider the pixel size differences and the radiance average has to be a pixel-size weighted average.

If my understanding is correct, then the global ADM not only rely on composite product's scene identification, CERES ADM for each pixel, but also on EPIC's radiances measurements (which rely on CERES-MODIS collocation and narrowband to broadband conversion) to derive the global mean ADM. The EPIC-based sunlit global SW flux (Su et al. 2018) has used EPIC radiances and CERES ADMs and does not really need global ADM and thus global ADM is essentially untested. From EPIC radiance to flux, it relies on CERES derived narrowband-broadband conversion and CERES ADM, therefore the EPIC global flux provides some consistent check but not absolute validation in my opinion.

Eq. 13 and 14. From these equations, we know that the NISTAR flux depends on unfiltered radiances from NISTAR and the global ADM derived from EPIC (which itself depend on many other instruments and procedures). I would strongly suggest the authors examine the global ADM and NISTAR's radiance measurements separately to understand the variability and trends from each of these components. The computation of global ADM can be refined as mean radiance could be computed with pixel-size weighted average. The NISTAR total radiance and NIR radiance are also worth looking at especially when LW is derived from total subtract the SW.

---

## Author Comment (AC1) · 22 Oct 2019

11. In ERB calibration your definition of filtered radiance as IRRADIANCE/ SOLIDANGLE is only true if the instrument has a completely flat spectral response, which from fig 2 is certainly not the case for NISTAR SW & NIR channels.

*We are not sure what the reviewer means here. NISTAR is a broadband instrument, it measures the energy from the spectral ranges defined in Fig. 2. The relationship between radiance and irradiance should not change with the spectral response function.*

113-132. This is based on CERES unfiltering of Loeb et al 2001 I assume (where they are also labelled Eqns 3 & 4 at https://journals.ametsoc.org/doi/pdf/10.1175/1520-0450%282001%29040%3C0822%3ADOURFT%3E2.0.CO%3B2 ). Is it completely identical to CERES using the same decades old CERES radiative transfer database? This might be important to briefly mention as it could help eliminate mere inversion biases when you compare to CERES later in the paper.

*The concept of unfiltering used here is the same as that by Loeb et al. (2001), but the database used here is different and was calculated specifically for this study. The current database contains 722 clear-sky cases and 1519 cloudy-sky cases (line 166), whereas the total number of cases used by Loeb et al. (2001) was 272. This information is added to the manuscript.*

142. This is confusing, although I accept it probably amounts to same thing, are look up tables of a & b values (Eqns 3 & 4) or a table of kappa ratios actually used? Only if both techniques are used separately should there be 4 rather than just 2 eqns?

*Thank you for catching this. The Equations 3 and 4 are the original method used to unfilter the CERES observations. As you know, we have scene-type information and Sun-viewing geometry for each CERES footprint, thus the regression can be applied based upon the scene type and Sun-viewing geometry of the CERES footprint. NISTAR views the entire Earth as a single pixel, and the cloud fraction, cloud type, and land/ocean portions differ from time to time. Luckily, the NISTAR SW spectral response function is such that the ratio between filtered and unfiltered radiances exhibit very little sensitivity to the scene types and Sun-viewing geometry. We rewrote the section on page 7 and 8 to correct this.*

143. How are spectrally dependent changes to the transmission of the quartz filter due to outgassing contamination measured after launch and throughout the mission?

*On-orbit measurements indicated that the filters have not degraded significantly since they were measured on the ground during calibration. On orbit measurements of the broadband transmission of the filter stack are continually made every three months using the earth as a source and the photodiode as a detector. The ratios of the on-orbit transmittances amongst each*

*of the two sets of 3 nominally identical filters of each type (SW and NIR) are within 0.2% of each other (as expected from ground measurements) and have remained stable to less than 0.1% throughout the mission. In the case of the SW filter (quartz), the on-orbit broadband transmittance is within 1% of the spectral transmittance of the filter stack over the wavelength range from 500 nm to 2500 nm.*

144. What about quartz filter leakage? Are you using the NIR channel for that somehow (similar to Loeb et al 2001 above)?

*A thin quartz filter can transmit significantly at wavelengths greater than many tens of micrometers, however, the NISTAR filter stacks consists of a pair of 3 mm thick quartz substrates—one is a bare uncoated substrate and the other has dielectric coatings to block light below about 700nm. At 3 mm thickness per substrate the transmittance below 100 micrometers is negligible. Loeb et al (2001) did not use any NIR channel.*

146. Are you sure no unfiltering of the total channel is required, if so how? Was its spectral response measured to be certain? How are you certain no changes to the effective gains of the cavity channels due to electronics radiation exposure are occurring? I'm assuming you do not have onboard blackbodies?

*We do not unfilter the total channel. The total channel spectral response is determined by the spectral absorptance of its cavity absorber, which, like a blackbody, relies on multiple reflections to achieve a high degree of absorptance (emissivity). Each cavity is conical in shape to trap light and is painted with a specular black paint, Z302, which has a very small component of diffuse reflectance. Measurements of the cavity absorptance made on the ground at wavelengths of 488 nm, 514 nm and 632 nm confirmed that the cavity absorbed more than 0.9997 of the incident light. Given the known spectral reflectance of Z302 to long wavelengths and the cavity design (verified at visible wavelengths), un-filtering of the total channel is not required.*

*The only electronics that affect the cavity channel gains are those that measure the electric power applied to the cavity heaters. Those electronics were chosen for their radiation tolerance and long term stability. Given the on-orbit radiation exposure levels, such degradation is not expected to significantly affect heater power measurements. Similar techniques and electronics are used to measure the total solar irradiance from space with a stability of less than 0.1%, which is sufficient to resolve the 11 year solar cycle. Unlike those measurements, degradation from UV exposure is not an issue here. You are correct, there aren't any on-board blackbodies to use as a references. Such blackbodies would have to have phase transition temperature references to be less sensitive to radiation exposure than the radiometers. This is because electronic temperature measurements are much more challenging than measurement of the power applied to the cavity heaters.*

147. How are the SW and Total channels balanced in the solar region as in Kratz et al 2002? (https://agupubs.onlinelibrary.wiley.com/doi/epdf/10.1029/2001JD001170)

*Since the NISTAR instrument only views the sunlit side of the Earth, there are no measurements taking during nighttime that can be used in the same manner as Kratz et. al. (2002), in which they looked at the correlation between nighttime total channel and window channel.*

261. what is a shutter cycle? Why is a boxcar filter used in the demodulation algorithm? Are details of these processes important?

*NISTAR utilizes a shutter to modulate light from the Earth just as a chopper wheel is used in the laboratory to modulate a light source. The shutter is opened and closed continuously with a 50% duty cycle with a period of nominally 4 minutes. Each 4 minute period is a shutter cycle. The demodulation algorithm is analogous to what is performed in a digital lock-in amplifier. Use of a boxcar filter having the width of a shutter period strongly rejects higher harmonics of the shutter frequency. Other low pass filters could be used. Note that additional filtering at lower frequencies, e.g., 4 hour running averages, are used to further reduce noise levels. Description is added on page 5.*

264. Recommended based on what, the URL does not work?

*Based on the noise level. The URL was temporally unavailable due to internal web maintenance, it should be available now. Sorry about that.*

266. Why 4 hour "running means" and how is this different from a 4-hour wide boxcar filter from the terminology you used earlier? Does this mean a 4-hour running mean is taken of the boxcar filtered then 2-hour averaged data? Why are the 4 hour means suggested by the NISTAR instrument team?

 *A running mean of 4 hours is conceptually the same as a boxcar filter. The 4 hour averages are additional filtering that occurs after the 4 minute wide boxcar filter to reduce noise levels. A four hour compromise is proposed as a trade-off between reducing noise and attenuating the signal of interest, however, the data is also provided without the additional filtering so the user may apply their own filter.*

286. These GERB comparisons need a reference.

*Reference "Doelling et al. (2013)" was provided on line 342, immediately after summarizing the comparison results.*

311. Why does the onboard data processing cause this?

*We removed this sentence in the revised version.*

315. How are the offsets countered, space looks?

*Yes. The shutter removes some, but not all offsets. Those that remain are removed with monthly space looks. Description is added on page 5.*

332. With as few as 10 EPIC results per day are these always equally spaced in time? If not, could this not lead to biases?

*When EPIC is in normal operations, it receives about 10 images daily during the winter cadence. They are normally spaced about 2 hours apart. EPIC receives about 20 images a day during the summer cadence and they are about 1 hour apart. If we simply compare the daily mean fluxes averaged using the EPIC image times with those averaged over the 24 hours, that would lead to biases. In this study, we only averaged the CERES SYN1deg using the hours that coincide with the EPIC times (line 376). Thus ensure both daily means are calculated using same number of hours.*

338. So it seems the LW difference is greatest in Northern Hemisphere Winter, when more ocean is observed? This may be a calibration artifact or error in knowledge of the NISTAR SW channel for the UV region. As per the point above for line 147, how are you balancing SW and Total channels to assure accurate LW in daylight?

*Preliminary analysis of the 2018 measurements does not show the same difference pattern (i.e. larger difference over the boreal summer months than the winter months), thus not supporting the hypothesis of the reviewer. As we mentioned earlier, NISTAR only views the sunlit side of the Earth and the same method used by Kratz et al (2002) cannot be applied here.*

352. "A comprehensive spectral database has been developed", so is it different from that used by CERES?

*Yes, and more details are added on page 7-8.*

355. So is a constant of 0.8690 used for all NISTAR unfiltering? Unfiltering of LEO scenes varies greatly by several percent especially for ocean scenes etc. So, it seems a value of 0.3% difference for primarily land vs Pacific Ocean scenes would vary more (and maybe adds to your seasonal cycle). What results lead to the 0.3% conclusion and did you try a scene by scene unfiltering?

*Based on the simulated filtered and unfiltered radiances for 722 clear-sky cases and 1519 cloudy-sky cases for each Sun-viewing geometry, the ratio between filtered and unfiltered radiances is extremely stable (see Table 2). Table 2 summarized the ratios and their standard deviation for each solar zenith angle bin for each scene type. For clear-sky case, each solar zenith angle bin contains over 57,000 simulations; and for cloudy-sky case, each solar zenith angle bin contains over 120,000 simulations. The largest ratio difference over different scene types happens under overhead sun, where the ratio for clear ocean is 0.8659 and is 0.8694 for clear land. Using constant unfiltering ratio of 0.8690, it could cause up to 0.3% unfiltering uncertainty if a clear ocean scene is encountered. However, NISTAR views the sunlit side of the Earth as a single pixel. There are always clouds and land mixed in. Thus we state the unfiltering*

*uncertainty should be less than 0.3%. We rewrote the unfiltering portion of the paper on page 7 and 8 to clarify the reviewer's concerns.*

370. Is this the PSF of the EPIC telescope separate from its array of detectors? How was it measured?

*We are not sure we understand the reviewer's question.  The PSF tells us where does the light measured in one pixel come from. It's a function of the instrument's entire optical system, telescope and detector.  The EPIC PSF was measured in the laboratory before launch, nominal PSF is given in Khlopenkov et al. (2017, SPIE).*

388. Again, this could be due to a constant unfiltering factor?

*Please see response above.*

392. Loeb et al 2018 only quotes the 1% accuracy figure as do you, please provide a peer reviewed SI traceable reference.

*The following CERES calibration references are added:*

*J. M. McCarthy, H. Bitting, T. A. Evert, M. E. Frink, T. R. Hedman, P. Skaguchi, and M. folkman. A summary of the performance and long-term stability of the pre-launch radiometric calibration facility for the Clouds and the Earth's Radiant Energy System (CERES) instruments. In 2011 IEEE International Geoscience and Remote Sensing Symposium, pages 1009–1012, 2011.*

*K. J. Priestley, G. L. Smith, S. Thomas, D. Cooper, R. B. Lee, D. Walikainen, P. Hess, Z. P. Szewczyk, and R. Wilson. Radiometric performance of the CERES Earth radiation budget climate record sensors on the EOS Aqua and Terra spacecraft through April 2007. J. Atmos. Oceanic Technol., 28:3–21, 2011.*

396. Please give a peer reviewed reference for the 2.1% NISTAR SW accuracy figure.

*NISTAR is a relatively new instrument and so far no peer reviewed publication describing the calibration is available. The presentation describing the NISTAR calibration is available at: https://avdc.gsfc.nasa.gov/pub/DSCOVR/Science_Team_Meeting_Sept_2019/L1/NISTAR_Godda rd%20Science%20Team%2020190917.pdf*

404. With so many error sources not well known it is wrong to simply add them all in quadrature, which assumes they are all random and independent. A more sophisticated error analysis is needed.

*The reviewer is correct that the error sources considered here were simply added to approximate the uncertainty. I would say this is a simplified estimate of the uncertainty, but not the wrong estimate. We know the sources of the uncertainty, but don't know the correlation of all the error sources and therefore unable to estimate the covariances of the sources considered here. The*

*uncertainty given here can be regarded as the upper bound, and this method has been used by Loeb et al. (2009) and Loeb et al. (2018).*

407. The 1.8Wmˆ2 accuracy for CERES LW applies for nighttime LW only. During the day which is always the case for NISTAR it is less accurate. This is because it requires the earlier discussed balancing of the SW and Total channel which if done wrong can result in measuring the Earth warmer at night than during the day for example (see Fig11b, Page 14 at https://journals.ametsoc.org/doi/pdf/10.1175/2010JTECHA1521.1). Hence for NISTAR which only views day LW, this is an important consideration.

*The reviewer is correct that the accuracy of the daytime and nighttime LW is different. The daytime LW uncertainty due to calibration is 2.5 Wm-2 (1 sigma). The combined uncertainty is updated based on the daytime LW flux uncertainty (line 461).*

415. Guesstimate? This is most unsatisfactory for any science paper, let alone one on climate measurements. Please do better.

*Changed to "estimate".*

423. Again, adding in quadrature for so many uncertain, often modelling terms is not acceptable. For example, consider how the error in knowledge of SW vs Total solar response could be systematic because of an error in the ground lab, it will partly cancel in the Total – SW subtraction.

*Please see our response above regarding the uncertainty estimation. The daytime LW flux uncertainty due to calibration is estimated by accounting for the calibration uncertainty in both total channel and SW channel, and the correlations between these two channels.*

428. This is true, in addition to the above-mentioned systematic nature of SW and Total errors not considered in your quadrature additions. A more sophisticated analysis is needed.

*As we stated above, the error analysis considered both SW and total channel. However, changes in error analysis won't affect the correlation between the LW flux from CERES and from NISTAR.*

Overall this paper has merit but needs work to fill in the blanks on some of the processes/references used. The large differences of NISTAR from CERES appears strange and would seem at first look to be largely from algorithm errors. I feel this could be acceptable being a new measurement, but needs to be stated more clearly in the paper as such.

*The NISTAR instrument is the first ever cavity radiometer placed at the L-1 point to measure the Earth's radiation. EPIC on board the DSCOVR also provides 10 narrowband observations from the same Sun-viewing geometry and the visible channels of EPIC are calibration against MODIS. When the global SW anisotropic factors were applied to the EPIC broadband radiance*

*(derived by applying narrowband-to-broadband regressions to EPIC blue, green, and red measurements), the EPIC SW flux agrees with the CERES SYN SW flux to within 2%. The good agreement indicates that the algorithm that we developed is accurate and is not the cause for the large discrepancy between NISTAR and CERES SYN. Even though there are discrepancies between the NISTAR fluxes and CERES SYN fluxes, we feel it is important to document the measurement, the algorithm, and the validation for future reference.*

The use of constant SW unfiltering also raises concern
and leads to the possibility it is a cause of the larger than expected seasonal
cycles, but more investigation is needed. Also some insight in the introduction
into the purpose of NISTAR would be good, such as giving illustration if and
how it complements the climate observing system discussed by Weilicki et al 2013
(https://journals.ametsoc.org/doi/pdf/10.1175/BAMS-D-12-00149.1 ).
In summary, this paper could become suitable for publication, given more work, research
and additions that address the points above. It should then be re-considered
under peer review.

*Based on the simulated filtered and unfiltered radiances for 722 clear-sky cases and 1519 cloudy-sky cases for each Sun-viewing geometry, the ratio between filtered and unfiltered radiances is extremely stable (see Table 2). Table 2 summarized the ratios and their standard deviation for each solar zenith angle bin for each scene type. For clear-sky case, each solar zenith angle bin contains over 57,000 simulations; and for cloudy-sky case, each solar zenith angle bin contains over 120,000 simulations. The largest ratio difference over different scene types happens under overhead sun, where the ratio for clear ocean is 0.8659 and is 0.8694 for clear land. Using constant unfiltering ratio of 0.8690, it could cause up to 0.3% unfiltering uncertainty if a clear ocean scene is encountered. However, NISTAR views the sunlit side of the Earth as a single pixel. There are always clouds and land mixed in. Thus we state the unfiltering uncertainty should be less than 0.3%. We rewrote the unfiltering portion of the paper on page 7 and 8 to clarify the reviewer's concerns.*

---

## Author Comment (AC2) · 22 Oct 2019

General comments:
This paper presents the scheme and algorithm of deriving TOA SW/LW flux from NISTAR measurements and comparison also made with the corresponding results derived from CERES. I am impressed by the detailed and clear description of the algorithms. The paper is very well written and relevant to the community. I recommend publication after addressing the minor issues listed bellowed. It doesn't seem that the uncertainties in the algorithms would give a consistent bias seeing in the differences between NISTAR and CERES. Has there been analysis with the NISTAR instrument measurements and calibration? The low correlation between NISTAR LW flux and that of CERES is puzzling. To bypass the potential uncertainties in part of the algorithms, it may be useful to look at the correlation between the NISTAR LW radiances and the CERES flux to see if they are correlated at all.

*The NISTAR instrument team (who produces the L1 data) is responsible for the instrument calibration and the team has presented their calibration at the DSCOVR science team meetings(https://avdc.gsfc.nasa.gov/pub/DSCOVR/Science_Team_Meeting_Sept_2019/L1/NIST AR_Goddard%20Science%20Team%2020190917.pdf). So far their analysis are mainly focused on the SW channel. NISTAR has three broadband electrical substitution radiometers (ESRs). All ESRs have a large background noise as they measure the change in incident optical power. Two steps are utilized to remove the background noise: first using a shutter to modulate the source which removes most of the background noise then using dark space view to remove the residual shutter-modulated background. The shutter modulated background is largest for the total channel and is smaller for the SW channel. As the LW is derived from the difference between total and SW channels, both total channel and SW channel background noises contribute to the LW uncertainty. The NISTAR total channel uncertainty is 1.5% and the SW channel uncertainty is 2.1%. Assuming the SW flux is 210 Wm-2 and the LW flux is 240 Wm-2, thus gives the total flux uncertainty as 450\*1.5%=6.8Wm-2, and the SW flux uncertainty as 210\*2.1%=4.4Wm-2. The resulted uncertainty in LW flux is 8.1 Wm-2, which can explain most of the LW differences between NISTAR and CERES SYN shown in Table 4. See added description on page 6. The low correlation is also caused by the background noise in both the total and SW channels. Details on NISTAR calibration are added on pages 5 and 6. Below is an example of August 2017, where the correlation between CERES SYN LW and NISTAR LW flux is about 0.38, and the correlation between CERES SYN LW flux and the NISTAR LW radiance is about 0.41. It is obvious that the low correlation is mainly from the instrument calibration.*

[Figure]

Specific comments:

Page 5 and 6: The authors have derived the regression equations for the unfiltered radiances (Eq 3 and 4); what is the reason for using the less accurate ratio method (Eq. 5 and 6)?

*The Equations 3 and 4 are the original method we planned to use for the NISTAR unfiltering. But unlike other LEO instruments that have scene-type information and Sun-viewing geometry for each footprint, and the regression can be applied based upon the scene type and Sun-viewing geometry of each footprint. NISTAR views the entire Earth as a single pixel, and the cloud fraction, cloud type, and land/ocean portions differ from time to time. Luckily, the NISTAR SW spectral response function is such that the ratio between filtered and unfiltered radiances exhibit very little sensitivity to the scene types and Sun-viewing geometry. We rewrote the sections on page 7 and 8 to correct this.*

Page 14: How are the portion of the Earth not visible to NISTAR decided? Also, similar to NISTAR missing some of the daytime portion of the Earth, it must be seeing part of the night time side of the Earth. Are these taking into account for the longwave calculations?

*The mask is calculated based upon the solar zenith angle and the EPIC viewing zenith angle and each EPIC pixel is identified as nighttime hidden to EPIC, or nighttime visible to EPIC, or daytime hidden to EPIC, or daytime visible to EPIC. Both the daytime and nighttime visible to EPIC are considered for the CERES SYN product to compare with the NISTAR LW measurements. Some clarification is added on page 14 and Figure 6b) is modified accordingly.*

---

## Author Comment (AC3) · 22 Oct 2019

Review on "*Determining the Daytime Earth Radiative Flux from National Institute of Standards and Technology Advanced Radiometer (NISTAR) Measurements*" by Su et al.

This paper documents the methodology to derive the broadband radiative flux from the measurements of the NISTAR instrument onboard of the DSCOVR mission. Some preliminary results based on this method are compared with the well-developed CERES data. The SW fluxes derived from the NISTAR compares reasonably well with CERES, but the LW fluxes from NISTAR have a systematic bias and low correlation coefficient when benchmarked with CERES.

The topic of this paper is important and suitable for AMT. The paper is well organized. However, the paper lacks some important technical details about the instrument and the methodology, as well as the author's opinion about the usefulness of the NISTAR product. In my view, some significant revisions are needed before the paper can be accepted for publication. Below is a list of questions and concerns I have.

1) The parameterization scheme described in Section 2 to obtain unfiltered radiance from observed filtered radiance is confusing. Up to line 132, the method seems to be based on the polynomial parameterization scheme in Eqs (3) and (4). But then it suddenly changed to the simply ratio-based parameterization in Eqs. (5) and (6). Why are there two types of parameterization? Which one is used?

*The Equations 3 and 4 are the original method we planned to use for the NISTAR unfiltering. But unlike other LEO instruments that have scene-type information and Sun-viewing geometry for each footprint, and the regression can be applied based upon the scene type and Sun-viewing geometry of each footprint. NISTAR views the entire Earth as a single pixel, and the cloud fraction, cloud type, and land/ocean portions differ from time to time. Luckily, the NISTAR SW spectral response function is such that the ratio between filtered and unfiltered radiances exhibit very little sensitivity to the scene types and Sun-viewing geometry. We rewrote the sections on page 7 and 8 to correct this.*

2) What is the FOV size of the NISTAR instrument? Does it observe the earth pixel by pixel (similar to EPIC) or as a whole? Does its FOV include some cosmic background and, if so, how is that treated?

*NISTAR observes the entire sunlit side of the Earth as one pixel. We specifically mentioned this on lines 53-54.*

3) Within its FOV, does the NISTAR instrument response to the radiance from different locations and angles equally? In other words, do the radiances from the edge of the earth disc have the same weighting as those from the center of the disc?

*Yes, NISTAR response to the radiance from different locations and angles equally. Optically the instrument is very simple—there aren't any lenses or mirrors, just filters, and a pair of apertures, and incident light is nearly perpendicular to the filters and apertures. The Earth subtends an angle of less than 1 degree from DSCOVR.*

*4)* It is stated that "*The biases in the anisotropy correction for the DSCOVR scattering angle are mitigated and potentially minimized by the wide range of different scene 71 types viewed in a given NISTAR measurement.*" Some references are needed to support it.

*We referenced the Su et al. (2018) paper here. As this is a very new way to measure the Earth radiative flux, no other references are available.*

5) In Su et al. (2018), a similar method is used to derive the fluxes from EPIC measurements. One of the byproduct from this EPIC-based method is the "global day-time mean SW radiance" $I\overline{\overline{bb}}$. Is it something directly comparable to the observation of NISTAR instrument? If so, some comparisons should be made because both EPIC and NISTAR have the similar sun-satellite geometry.

*Indeed, we have derived the "global daytime mean SW radiances from EPIC". They are consistently lower than the radiances from NISTAR. Below are the comparison between NISTAR and EPIC radiances for April and July 2017. The mean differences are between 4 to 6 Wm-2sr-1. We chose not to include these results in this paper to avoid any confusions and the EPIC and CERES comparisons were provided in Su et al. (2018).*

[Figure]

6) I have several questions about the method described in Section 3c. First of all, what is the theoretical based for Eqs 9~ 11? If my understanding is correct, the global mean SW flux is

$$F = \iint_{sunlit} \frac{I[\theta_0(r), \theta^e(r), \emptyset^e(r), \chi(r)]}{R(\theta_0, \theta^e, \emptyset^e, \chi)d^2r} d^2r$$

Where r denotes a point on earth. But this is not equal to

$$\frac{\iint_{sunlit} I[\theta_0(r),\theta^e(r),\phi^e(r),\chi(r)]d^2r}{\iint_{sunlit} R(\theta_0,\theta^e,\phi^e,\chi)d^2r}.$$

More detailed mathematically derivations are needed here. Secondly, one might ask if a global mean anisotropic factor is even physically meaningful? The average is over a large range of viewing angles and scene types. Does the result have any physical meaning? Moreover, are the angular and spectral averaging independent and can be treated independently? The derivations in Section 3c seem to suggest they are independent, but this is not obvious to me. Some clarification is needed.

*We agree with the reviewer that the above two equations are not the same. The first equation is how we calculate global mean flux from low-Earth orbit satellites (i.e. CERES) using the footprint level data (resolution on the order of 20 km) by first grid the data then area weight to calculate the global mean. We did not use the second equation in our study to derive the fluxes from NISTAR radiance measurements.*

*To derive the global mean flux from NISTAR measurements, a corresponding anisotropic factor to characterize the sunlit portion of the Earth as a whole is needed, and this is the definition of the global mean anisotropic factor we used in the paper. The global mean anisotropic factor is derived by using the radiances and fluxes defined in the CERES angular distribution models (ADMs). The global mean radiance and flux from CERES ADMs were calculated independently (see Equations 8 and 9 in the revised version). They are used to derive the global anisotropic factor (Equation 10) and subsequently to convert the NISTAR radiance to flux (Equation 11). The deviation of the NISTAR flux used here is not the same as illustrated by the second equation above. This method has been tested for both the NISTAR and EPIC measurements.*

7) This paper only shows "how to do it" but does not explain "why to do it" other than it can be done. I understand that this paper is to document the method used to derive the flux from the radiance observations of NISTAR. But I think in addition to the technical details the reader would apricate some insights and opinions from the authors about the usefulness of the product. We already have the state-of-the-art CERES flux product and in Su et al. (2018) flux product has also been developed. What is new/novel/important about the NISTAR flux product other than the fact it can be done? What kind of applications can this product be used for? Some discussions about these important questions should be added to the abstract and conclusion parts.

*We added some information on NISTAR measurement and its utility in the introduction (lines 59-68). We also added some perspective on the utility of NISTAR SW fluxes in the conclusion section (lines 486-492).*

---

## Author Comment (AC4) · 22 Oct 2019

Review of "Determining the Daytime Earth Radiative Flux from National Institute of Standards and Technology Advanced Radiometer (NISTAR) Measurements" by Su et al. 2018

General comments:

This manuscript derives sunlit side of the Earth's radiation budget (SW and LW) from a single pixel measurement of NISTAR instrument on board the DSCOVR mission and compares with the radiation fluxes derived from the CERES measurements. This is a very interesting and important work as the Earth's radiation budget has been so far solely measured by the ERBE/CERES project and there are very little independent and direct measurements of these important quantities. This work builds upon many previous works the team has been working on for many years including narrowband broadband conversion, ADM, GEO/LEO composite cloud products etc. The paper is well written and structured. I do have some questions and suggestions regarding the derivation of global ADM and evaluation of each components of the fluxes.

Specific comments:

Line 98: What is the uncertainty level of NISTAR L1B radiance? What kind of calibration procedures have been used to produce the L1B radiance? You have discussed some of the issues later in the paper but it's worthwhile to have a paragraph to discuss the NISTAR at the beginning of the paper. NISTAR provides a completely different methodology of estimating the earth's radiation budget and independent check of Earth's radiation budget created from CERES measurements, the difference found in this article is very serious and should be adequately explained. NISTAR's absolute calibration and uncertainty is of fundamental importance, otherwise the readers would question the well-established CERES products.

*The NISTAR instrument team (who produce the L1 data) is responsible for the instrument calibration and the team has presented their calibration at the DSCOVR science team meetings (https://avdc.gsfc.nasa.gov/pub/DSCOVR/Science_Team_Meeting_Sept_2019/L1/NISTAR_Goddard%20Science%20Team%2020190917.pdf). So far their analysis are mainly focused on the SW channel. NISTAR has three broadband electrical substitution radiometers (ESRs). All ESRs have a large background noise as they measure the change in incident optical power. Two steps are utilized to remove the background noise: first using a shutter to modulate the source which removes most of the background noise then using dark space view to remove the residual shutter-modulated background. The shutter modulated background is largest for the total channel and is smaller for the SW channel. As the LW is derived from the difference between total and SW channels, both total channel and SW channel background noises contribute to the LW uncertainty. The NISTAR total channel uncertainty is 1.5% and the SW channel uncertainty is 2.1%. More details on NISTAR calibration is added on page 4-6.*

Line 147. The conversion from filtered to unfiltered radiances used the ratio derived from model simulation data using eq 5 and 6. Why not using the regression (3) and (4)? The regression indicates the ratio could not be constant because it's a quadratic function and has an offset. It's justified to use a constant ratio between the two if the ratio varies little as for the SW band, but a constant ratio for NIR would introduce an unnecessary source of error (1_2%) for the NIR and I don't see why you should

abandon the regression.

*The Equations 3 and 4 are the original method we planned to use for the NISTAR unfiltering. But unlike other LEO instruments that have scene-type information and Sun-viewing geometry for each footprint, and the regression can be applied based upon the scene type and Sun-viewing geometry of each footprint. NISTAR views the entire Earth as a single pixel, and the cloud fraction, cloud type, and land/ocean portions differ from time to time. Luckily, the NISTAR SW spectral response function is such that the ratio between filtered and unfiltered radiances exhibit very little sensitivity to the scene types and Sun-viewing geometry. As we don't have the scene-type information, the regression method can't be applied to NIR either. We rewrote the sections on pages 7 and 8 to correct this.*

Line 152: Did you use NIR in this work? If not, could you explain why NISTAR takes the NIR measurement?

*We did not use the NIR channel in our work. When NISTAR was design in the 1990s, the primary utility for the NIR channel is to study the enhanced absorption of SW radiation by clouds (Collins 1998) and more recently by Carlson et al. (2019) to look at the spectral ratio of the sunlit side of the Earth and the potential of using the ratio for model evaluation (see introduction on page 3).*

Line 187: EPIC images have 8x8 km2 resolution at nadir and are 1/cos(vza) larger at larger view zenith angles. The EPIC cloud products are retrieved at its native resolution with (2014x2014) pixels in a granule. Some channels have degraded into 1024x1024 for downlink but reversed to 2014x2014 afterwards.
Equation (9) and (11), Ij and Fj seem to refer to radiance and flux in each EPIC composite pixel. Do you actually use those in the mean ADM calculations? If yes, did you use the EPIC measured narrowband radiances to compute the broadband radiance and flux for each pixel?

*In Equations (9) to (11) (now Equations 7-9), the ^I and ^F refer to the radiances and fluxes from the CERES ADMs (same symbols are used in Equation 6). To clarify the confusion, we added a sentence on page 12 (lines 291-292). They are not from EPIC.*

Why did you grid the fluxes into 1x1 grid boxes and not the radiances? The global mean flux is computed from Eq. 11 to take care of different sizes of grids in each latitude. If you grid the radiance, then you would compute the mean radiance the same fashion as the flux. Otherwise, if you average the radiance from each pixel directly, then you would also have to consider the pixel size differences and the radiance average has to be a pixel-size weighted average.

*Radiance and flux are fundamentally different physical quantities. Radiance is the total amount of energy confined to a given direction per unit surface area (in Wm-2sr-1). One essential property of radiance is that it is additive, meaning if several sources contribute to the radiance at a particular point and in a particular direction, the total radiance is the sum of the radiances from each source as if it were acting alone (Bohren and Clothiaux, 2006). On the other hand,*

*flux is the energy per unit surface area (Wm-2) and need to be area-weighted when compute global means.*

If my understanding is correct, then the global ADM not only rely on composite product's scene identification, CERES ADM for each pixel, but also on EPIC's radiances measurements (which rely on CERES-MODIS collocation and narrowband to broadband conversion) to derive the global mean ADM. The EPIC-based sunlit global SW flux (Su et al. 2018) has used EPIC radiances and CERES ADMs and does not really need global ADM and thus global ADM is essentially untested. From EPIC radiance to flux, it relies on CERES derived narrowband-broadband conversion and CERES ADM, therefore the EPIC global flux provides some consistent check but not absolute validation in my opinion.

*The global ADM is derived using the scene identifications (surface type, cloud fraction, cloud optical depth, etc. ) from EPIC composite to select the anisotropic factors from the CERES ADMs. EPIC radiances are not used.*

*In Su et al. (2018), we used the same methodology to derive the global SW anisotropy factors (same as Equations 7-10 in the revised version). They are then applied to the EPIC global daytime mean "broadband" SW radiances, which were derived by using narrowband-to-broadband regressions. The EPIC global daytime mean "broadband" SW radiances are analogous to the NISTAR SW measurements, and the same SW anisotropy factors were applied to both NISTAR and EPIC SW radiances to derive SW fluxes. As demonstrated in Su et al. (2018), the SW flux from EPIC agree with CERES SYN to within 2%, which means that the method that we developed to derived the global mean anisotropic factors are robust.*

Eq. 13 and 14. From these equations, we know that the NISTAR flux depends on unfiltered radiances from NISTAR and the global ADM derived from EPIC (which itself depend on many other instruments and procedures). I would strongly suggest the authors examine the global ADM and NISTAR's radiance measurements separately to understand the variability and trends from each of these components. The computation of global ADM can be refined as mean radiance could be computed with pixel-size weighted average. The NISTAR total radiance and NIR radiance are also worth looking at especially when LW is derived from total subtract the SW.

*Again, EPIC composite provides the cloud properties that needed to select the anisotropy factors. We don't use any EPIC measurements in this study. The global mean anisotropy factors are calculated by deriving the anisotropic factors for each EPIC pixel. We did examine the NISTAR radiance against the "global daytime mean SW radiances from EPIC", derived by using narrowband-to-broadband regressions (Su et al. 2018). The NISTAR radiances are consistently greater than the EPIC "broadband SW" radiances. Below are the comparison between NISTAR and EPIC radiances for April and July 2017. The mean differences are between 4 to 6 Wm-2sr-1. We chose not to include these results in this paper to avoid any confusions and the EPIC and CERES comparisons were provided in Su et al. (2018).*

[Figure]

As noted by the reviewer, the LW radiance is derived by subtracting SW from the total. Thus the LW contains information of the total channel. As the focus of this paper is to derive SW and LW fluxes from NISTAR and validate the product with CERES product, and there aren't any global daytime total and NIR measurements that can be used to compare with the NISTAR measurements, simply looking at the NISTAR total and NIR channel measurement won't add value to the paper.

---

## Author Response (AR2)

Review of "Determining the Daytime Earth Radiative Flux from National Institute of Standards and Technology Advanced Radiometer (NISTAR) Measurements" by Su et al. 2018

The authors have addressed most of the reviewers' comments adequately, I only have a couple of minor comments:

*Thank you for spending time to review the paper!*

1) The clear sky radiance database includes many surfaces, aerosol types and optical thickness, what atmosphere profiles do you use? Are there any variations in terms of humidity, pressure, traces gases?

*For most cases, we only used one atmospheric profile (standard), we rerun the clear ocean case using the tropic profile. The ratios show very little sensitivity to the profiles used, only affecting the ratio to the fourth decimal point. We added a sentence on page 7 (line 179-181) to state this fact. We also realized that we only included one aerosol type in our calculation and also tested the sensitivity to aerosol types over clear ocean and found very little sensitivity.*

2) A 4-hour running mean is applied to the NISTAR L1B radiances but the EPIC based global anisotropy is instantaneous. How much global mean anisotropy varies during the day and season? Would applying a 4-hour mean anisotropy or applying anisotropy to the NISTAR radiances before running mean make any differences?

*To reduce the noise level of the NISTAR measurements, we were advised to use the running means of NISTAR radiance. To answer the reviewer's questions, we redid the flux calculation by simply using hourly mean NISTAR radiances and tested the flux biases and RMS errors for June and July 2017. For June, the monthly mean NISTAR SW flux changed from 213.5 to 213.4 Wm-2, and the RMS error increased from 14.6 to 17.3 Wm-2; for July, the monthly mean NISTAR SW flux 209.2 to 208.5 Wm-2, and the RMS error increased from 16.0 to 16.9 Wm-2. For LW, monthly mean fluxes changed by less than 0.5 Wm-2 for both months, but the RMS errors increased by more than 4 Wm-2 for June and 6 Wm-2 for July. These results indicate that using running means do reduce the noise but has very little impact on monthly mean flux (less than 0.3%). The global mean anisotropic factors can change by up to 2-3% from hour-to-hour depending on the fraction of land and ocean in the EPIC field-of-view, and the viewing geometries of land and ocean, and the cloud properties.*

*We added a couple of sentences on page 16 (line 388-392).*

3) L8-9, You may want to use sunlit consistently through the paper.

*Modified, thank you!*